**SOFTWARE**

# Gramtools enables multiscale variation analysis with genome graphs

Brice Letcher[1*], Martin Hunt[1,2] and Zamin Iqbal[1*]

*Correspondence:
bletcher@ebi.ac.uk; zi@ebi.ac.uk
[1]EMBL-EBI, Hinxton, UK
Full list of author information is
available at the end of the article

## Abstract

Genome graphs allow very general representations of genetic variation; depending on the model and implementation, variation at different length-scales (single nucleotide polymorphisms (SNPs), structural variants) and on different sequence backgrounds can be incorporated with different levels of transparency. We implement a model which handles this multiscale variation and develop a JSON extension of VCF (jVCF) allowing for variant calls on multiple references, both implemented in our software `gramtools`. We find `gramtools` outperforms existing methods for genotyping SNPs overlapping large deletions in *M. tuberculosis* and is able to genotype on multiple alternate backgrounds in *P. falciparum*, revealing previously hidden recombination.

**Keywords:** Genome graph, Pangenome, Variant calling, *Plasmodium falciparum*, *Mycobacterium tuberculosis*, VCF

## Background

Variant calling, the detection of genetic variants from sequence data, is a fundamental process on which many other analyses rely. There are two standard approaches, each with their own limitations. For Illumina data, mapping to a reference genome causes reference biases that affect discovery and genotyping: mapped reads favour the reference allele and reads in divergent regions fail to map [1–4]. For PacBio/Oxford Nanopore Technology (ONT) data, genomes can be fully assembled, and therefore, the discovery and genotyping problems are in principle partially solved, by aligning each assembly against a reference. (There are caveats about how to get high per-base quality, either by hybrid ONT/Illumina or PacBio Hifi reads, but we leave this aside). However, the problem of how to coherently represent all of the variants in a cohort, comparing all against all, remains challenging both algorithmically and in terms of outputting results.

There are data structures that in principle can genotype alternate alleles which include both long structural variants and SNPs—some implementations include `Cortex`, `GraphTyper`, `vg`, and `BayesTyper` [4–7]. All of these are based on graph representations of one form or another ranging from genotyping a whole-genome de Bruijn graph

(`Cortex`), mapping all reads to a whole-genome Directed Acyclic Graph (DAG) of informative k-mers (`BayesTyper`), mapping all reads to a whole-genome graph of minimising k-mers and matched haplotype index (`vg/Giraffe` [8]) or remapping premapped reads either to local DAGs of SNPs and indels off the reference (`GraphTyper`) or to graphs built from structural variant breakpoints (`GraphTyper2` [9]). These all reduce the impact of reference bias, and allow cohort genotyping at consistent sites, but all of them struggle with the issue of representation. We highlight two important types of situations which pose representation challenges—any good solution will need to address these, and conversely, solving these alone would address most of the needs of most users.

First, a long deletion that "covers" 10 SNPs will in principle have $2^{10}$ alternate alleles. This would be painful to output in the widely used variant call format [10] (VCF), but more fundamentally, it would force the genotyper to make statements about long alternate alleles that only differ by SNPs. If two long alleles, one true and one not, differed by just one SNP, then under most models they would have very similar likelihoods, and it would be impossible to tell clearly which was the correct allele. In other words, there would be no fine-scale data about variants, making SNP filtering impossible (Additional File 1: Figure S1).

Indeed, there is no tool (to our knowledge) that self-advertises as supporting joint SNP and structural variant (large insertions/deletions) genotyping (although we note the software `Birdsuite` was developed for jointly genotyping SNPs and copy-number variants [11]). Benchmarking in this paper shows `GraphTyper2` and `vg` both work in this scenario if given VCF input. The second key challenge occurs when variants fall on different genetic backgrounds such as diverged MHC haplotypes [3] or large insertions—for this there is no current solution.

We highlighted above two key situations to address: SNPs as alternatives to long deletions, and SNPs on top of long alternate haplotypes. In both of these cases, variants are bound by relationships. In the first, they are mutually exclusive and in the second, they occur on top of alternative sequence backgrounds, thus combining exclusion and a hierarchy. We call such variation *nested* and identify these relations as sufficient to capture a valuable proportion of natural genetic variation. We therefore model the genome as a directed acyclic graph that is a succession of locally hierarchical subgraphs. This is a rich enough model to incorporate our key use-cases, without incurring the price of excessive generality; we discuss the benefits and limitations in the "Discussion" section. Based on this, we can use `gramtools` to identify these nested site relationships, leverage them during genotyping and output variants, genotypes and likelihoods in a file format extending VCF. This is, to our knowledge, the first framework for jointly analysing genetic variation at different scales (SNPs and structural variants) and on different sequence backgrounds.

We start by detailing the genome graph workflow implemented in `gramtools` and an algorithm for genotyping nested variation. We build graphs of variation from 2498 samples at four *Plasmodium falciparum* surface antigen genes which harbour high levels of diversity, including two that each have two diverged allelic forms (DBLMSP and DBLMSP2). We use simulated haplotypes from graphs to evaluate the impact of nesting on genotype confidences, and then benchmark with empirical data from 14 *P. falciparum* samples which have high quality PacBio assemblies for ground truth. We then address the two canonical use-cases we highlighted above. We show that `gramtools`

outperforms `vg` and `GraphTyper2` when genotyping long deletions and all the overlapping small variants from a cohort of 1017 *Mycobacterium tuberculosis* genomes. Finally, we apply `gramtools` to the use-case for which no current solution exists. We genotype 706 African and SE Asian *P. falciparum* genomes at the gene DBLMSP2, which possesses variation on two diverged backgrounds which had previously appeared to either never or, very rarely, recombine. This generates the first map of genetic variation on both diverged backgrounds, revealing patterns of recombination that were previously unknown.

## Results

`gramtools` implements a workflow for building, genotyping and augmenting genome graphs (Fig. 1). Genotyping serves two main use cases in this workflow. First, it is used for inferring a sample's closest path in the graph, which we can use as a "personalised" reference genome, since it should be a closer approximation than any individual genome would be. We are thereby able to discover new variants by using standard reference-based variant callers with this personalised reference, an approach previously described in [3, 12, 13]. Second, `gramtools` is used for genotyping cohorts of samples on a graph containing all such discovered variants. Neither case requires finding variants absent from the graph because novel variants are found by standard tools applied to the personalised reference. Thus, while other tools such as `vg` [4] perform full alignment of reads to the graph, `gramtools` achieves the same aims while only needing to do exact matching of reads

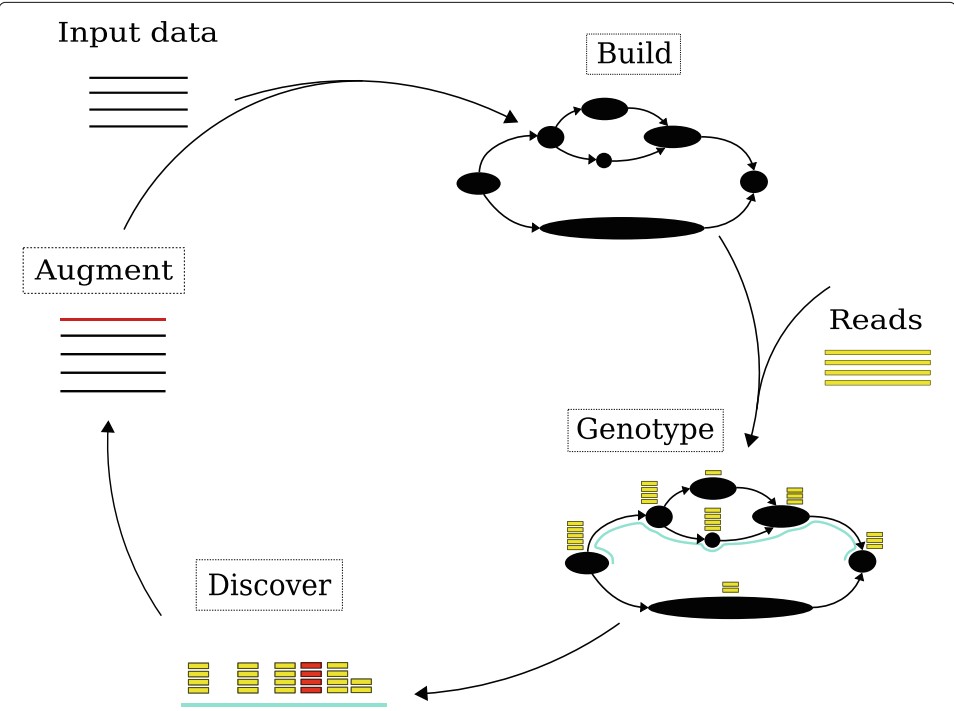

**Fig. 1** Genome graph workflow implemented in `gramtools`. Black nodes represent genomic sequence or site entry/exit points. Build consists of producing a genome graph that is directed and acyclic, a requirement for `gramtools`. Genotyping consists of calling alleles at each variant site and thereby inferring a haploid personalised reference genome for a sample. New variants, shown in red, are discovered using standard reference-based callers run against the personalised reference

(after quality trimming). Our implementation is currently limited to high-accuracy short reads (e.g. Illumina); support for long erroneous reads requires matching read substrings.

## Graph constraints and genotyping with the vBWT

In `gramtools`, sequence search in genome graphs is supported using the compressed suffix array [14] of a linearised representation of the graph, which we call variation-aware Burrows-Wheeler Transform (vBWT). The vBWT algorithm was introduced with a proof-of-concept implementation in [12]; details of how it converts BWT string matching to graph mapping are provided in the "vBWT data structure in `gramtools`" section. A key requirement of the vBWT is that the graph be decomposable into a succession of subgraphs (sites) each of which is strictly nested (see Fig. 2 and the "Graph definitions" section for formal definitions) interspersed by linear, non-variable regions.

This results in a graph where genotyping a site with alternate alleles is well-defined, preserving a notion with biological value, but also places some restrictions on the structure of the genome graph. We give an example in Additional File 1: Figure S3 showing a pair of allowed/disallowed graphs that generate equivalent sequence.

These graphs can be built from multiple sequence alignments (MSAs) or a reference genome plus VCF file; we used MSAs in this paper. The construction process, first introduced in [15], is explained further in the "Genome graph construction and `make_prg`" section, and we consider the implications of this model in the "Discussion" section. The original vBWT implementation was slow and did not support nesting [12]. In this paper, we introduce our nesting implementation and have optimised the codebase to improve mapping, coverage recording and genotyping.

## Genotyping nested genome graphs

`gramtools` genotypes a nested DAG in which variant sites have been defined (see the "Graph definitions" section). Sites are genotyped independently, choosing the maximum-likelihood allele under a coverage model that draws on ideas from `kallisto` [16],

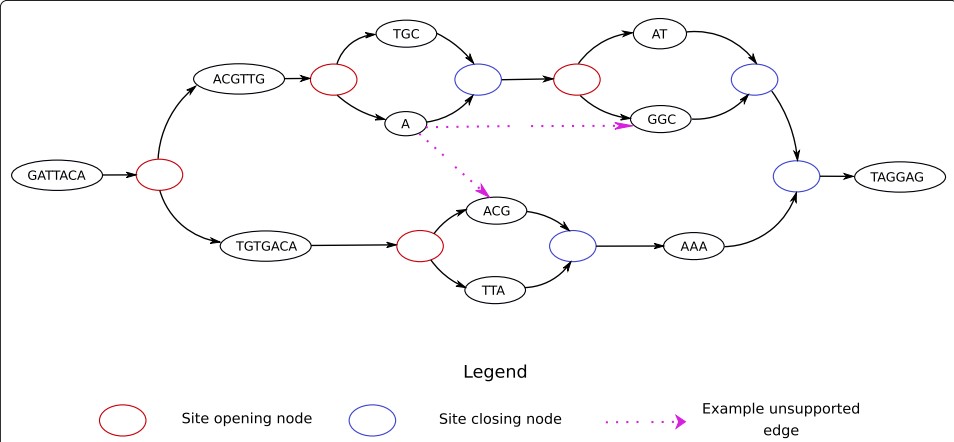

**Fig. 2** Gramtools requires variation to be expressed as a nested directed, acyclic graph (DAG). A nested DAG represents the genome as a DAG with a single source and sink, which can be decomposed into a succession of subgraphs (or sites). Each site starts with an opening node, and finishes with a closing node, and consists of strictly nested sub-sites. This allows hierarchical genotyping of alternate alleles. Strict nesting means that sub-sites must close off and complete before their parent site and without connecting to different sub-sites (e.g. the dotted pink edges would not be permitted)

including both per-base coverage information, and equivalence class counts for reads that could equally support different alleles (see the "Genotyping model" section). Genotype likelihoods are calculated for each allele at a site, and the ratio of likelihoods between the maximum likelihood allele and the next best is termed the genotype confidence. The personalised reference (PR) genome inferred by gramtools is the path obtained by taking the maximum-likelihood call at each variant site, and the genotype confidence at each site provides a measure of the adequacy of the inferred PR. For example, a stretch of low-confidence calls suggests no close path in the graph was found or that no reads mapped in this region. While the genotyping model can handle both haploid and diploid cases, in the diploid case two unphased PRs are produced (as was done in [3]) whereby the two alleles at heterozygous sites get randomly allocated to each. In this paper, we evaluate gramtools on haploid organisms only.

With nested variation, we apply the genotyping model recursively from child sites to their parents, with candidate alleles in parent sites generated based on the genotype calls of child sites (see the "Nested genotyping" section). We refer to each outgoing branch from a parent site as a *haplogroup*; if there is no nesting then these correspond to alleles, but when there is nesting, they can be seen as alternate sequence backgrounds (see Fig. 3 in the following section)

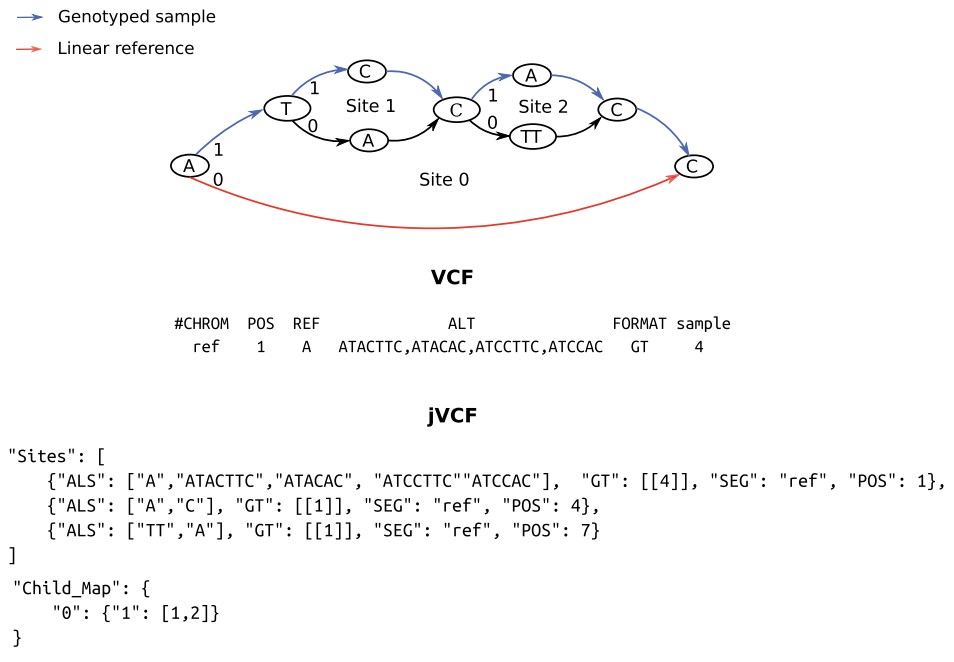

**Fig. 3** Example VCF and jVCF calls for a graph containing nested variation. Three sites exist in the graph, and site-opening edges are labelled by haplogroup (0 or 1). Below the graph, parts of VCF and jVCF files describe a genotyped sample (blue path in the graph). In VCF, only one variant site is genotyped, with the "sample" column stating that allele 4 has been called (this corresponds to "ATCCAC", as the numbering is 0-based). However, the two nested variants—sites 1 and 2—cannot be expressed independently from site 0, as they do not occur on the linear reference (red path in the graph). jVCF stores the same information as VCF in the "Sites" array (one entry per site) and additionally records site relationships using a "Child_Map" entry, which states that sites 1 and 2 occur under site 0 (first key), haplogroup 1 (second key). This places sites 1 and 2 on an alternate reference background (the sequence "ATACTTC") allowing them to be expressed in the "Sites" entry. Alternate reference sequences are obtained by following haplogroup 0 at each nested variant site (here this spells "ATACTTC")

### jVCF output format

In attempting to provide genotypes at all variant sites within a cohort, one is inevitably faced with densely clustered genetic variants, which leads to difficulties when using VCF. First, the genomic positions of output records can overlap, implying they should be considered jointly. Second, sites can occur on sequence backgrounds (haplogroups) that do not include the reference genome sequence. In VCF, overlapping records require careful genotyping and alternate references are not supported. To address these limitations, gramtools outputs a variant call file in JavaScript Object Notation (JSON), a widely used format for storing data as key-value pairs [17]. The format stores variant records mirroring VCF and additionally stores genome graph-specific information—we thus call it jVCF.

To illustrate the purpose of jVCF, we show in Fig. 3 a graph containing nested variation and parts of VCF and jVCF files describing genotype calls in this graph. Both formats store the reference and alternate alleles of a variant site and its location in the genome and can genotype the first site (site 0) in the graph. However, sites 1 and 2 occur on a different sequence background from the linear reference and thus cannot be expressed independently of site 0 in the VCF output. In jVCF, the parent/child site relationships are stored in a "Child_Map" entry which locates variant sites based on what sequence background (haplogroup) they fall on. This enables jVCF to express sites 1 and 2, by placing them on a different reference background with its own coordinates and to give independent genotypes for those sites. In jVCF, storing site relationships also makes incompatibilities between sites explicit, allowing variants in overlapping genomic positions to be consistently genotyped (see the "Nested genotyping" section).

A full format specification for jVCF is provided in Additional File 1. In addition to a jVCF file, gramtools outputs a regular VCF file containing only non-nested sites, yielding a VCF file with no overlapping records and referring only to the linear reference genome.

### Validation of nested genotyping with simulated data

Our first simulation was designed to evaluate both genotyping performance and whether nested genome graphs resulted in improved calibration of genotype confidences. We based the simulation on a real example where there are two alternate haplotypes each bearing variants, building graphs of *P. falciparum* variation for two genes, DBLMSP and DBLMSP2. These genes exhibit a dimorphism at the Duffy Binding-Like (DBL) domain, each having a region > 500bp in length with two allelic forms that are highly diverged [18, 19]. We built two versions of the graph, one without any nesting, and one allowing nesting up to five levels deep (see the "Genome graph construction and make_prg" section). The graphs were built from high confidence variant calls in 2498 samples from the Pf3k project [20] (see the "Graph construction" section for details). The graph without nesting contained 451/413 variant sites for DBLMSP/DBLMSP2 respectively and the graph with nesting contained 558/500 variant sites respectively. The nested graphs contain more sites because they allow SNPs/indels to occur on different sequence backgrounds, as illustrated in Fig. 4. We randomly sampled 10 paths from the non-nested graph (which therefore exist in the nested one), recorded the implied truth variant calls, simulated reads from the paths and passed them to gramtools for genotyping (see the "Path and read simulation"

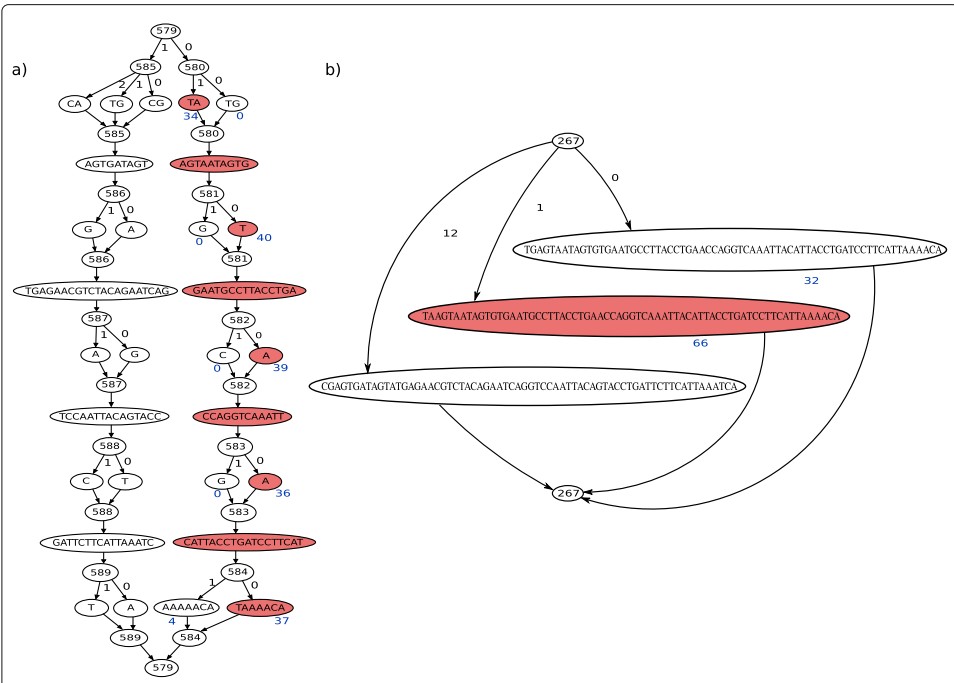

**Fig. 4** Nested graphs improve site resolution and coverage differences. A call for part of DBLMSP2 is shown (*P. falciparum* reference genome Pf3D7 chromosome 10 positions 1433921:1433987). Red nodes mark the called allele and spell the same sequence across nested (**a**) and non-nested (**b**) graphs. Numbered nodes mark variant sites and edges are labelled by haplogroup. Blue text under the nodes gives read coverage for the called and next best allele. In the non-nested graph, the next best allele is long and only one SNP away from the best allele so that reads mapping to common sequence add coverage to both. This reduces coverage differences compared to the nested graph. For clarity, only 3 of the 13 alleles that exist in the non-nested graph are shown in **b**

section for details). Out of 17,280 evaluated calls in the non-nested graph, gramtools recovered 99.9% (recall) and of all calls made, 99.8% were correct (precision). In the nested graph, out of 21,160 evaluated calls, recall and precision were both 99.9%.

While `gramtools` genotypes both types of graphs to high accuracy, nested graphs provide better call resolution. This is shown in Fig. 4 for (a subregion of) DBLMSP2 where the nested graph reflects the allelic dimorphism: SNPs/small variants fall on top of each allelic form. We also confirm that the genotype confidence of correct calls is also increased, since nesting allows likelihood calculations based on coverage precisely at SNPs on alternate haplotypes, rather than an average across the whole haplotype (coverage shown in Fig. 4, and effect on confidences shown in Additional File 1: Figure S6).

### Benchmarking `gramtools` genotyping against single-reference variant callers at surface antigens

Building on the simulation results in *P. falciparum*, we set out to evaluate `gramtools` genotyping in comparison with standard single-reference callers `SAMtools` [21] (the classical "pileup" variant caller) and `Cortex` [5] (which discovers bubbles in a de Bruijn graph, and then maps flanking sequence to the reference to get coordinates). We include `Cortex` because it has previously been shown to produce high-quality calls even in the indel-rich *P. falciparum* genome and can successfully identify alternate alleles at the genes

we analyse here [22] but has no capacity to consider nested variation. We built a whole-genome graph containing variation from 2498 *P. falciparum* samples (as in the simulation experiment) in four surface antigen genes: DBLMSP, DBLMSP2, EBA175, and AMA1 (see the "Graph construction" section). The additional genes, EBA175 and AMA1, are both vaccine targets, where there is great value in being able to correctly identify known and novel variation [23, 24]. We used 14 *P. falciparum* validation samples with both Illumina data and high-quality PacBio long-read assemblies [25], and which had been excluded from graph building, to assess `gramtools` genotyping. Performance is measured as the edit distance between the gene sequence with called variants applied and the long-read assembly, normalised by gene length. Note that the `gramtools` graphs are themselves built from `Cortex` calls in the 2498 samples (see the "Graph construction" section); thus, we are comparing genotyping via a graph of known population variation (`gramtools`) with reference-based variant calling (`SAMtools` and `Cortex`).

We show in Fig. 5 the scaled edit distance (i.e. edit distance divided by gene length) achieved by these tools on the 14 validation samples, aggregated across all four genes. As a baseline, we show the distribution of scaled edit distances between the 3D7 (reference) gene sequences and the truth assemblies, giving a mean distance of 3.7% (top-left panel, dotted line). `SAMtools` and `Cortex` both improve on this, achieving means of 2.3% and 1.3% respectively. `gramtools` outperforms both variant callers, achieving a mean of 0.6%. We provide performance for each individual gene in Additional File 1: Figure

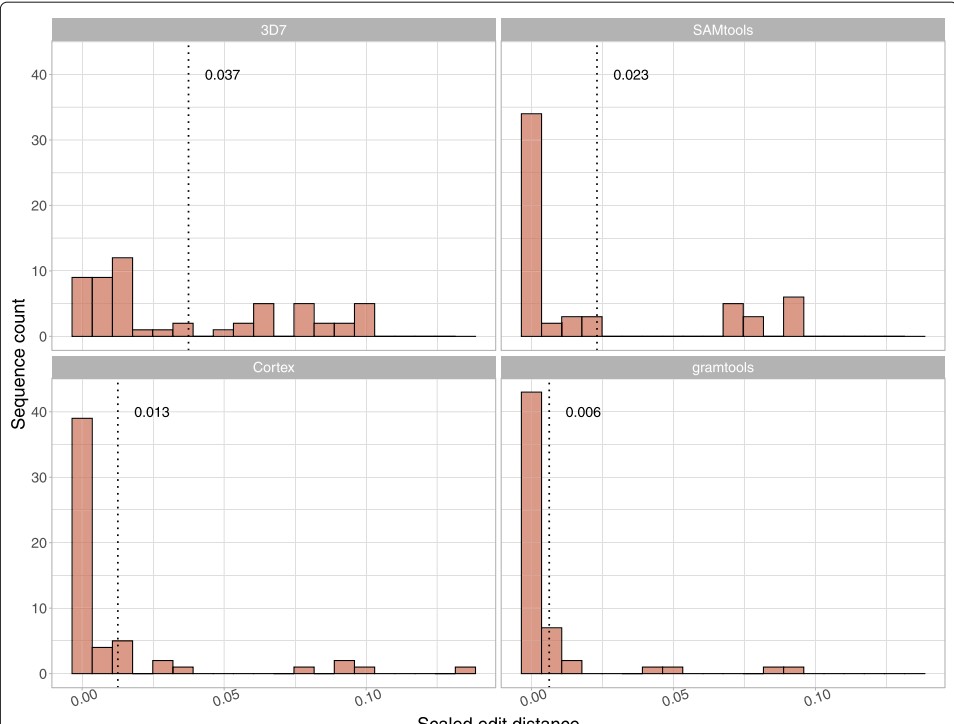

**Fig. 5** `gramtools` genotyping compared to reference-based callers at four surface antigens in 14 validation samples. The *x*-axis is the scaled edit distance (edit distance divided by the length of the gene) to the true sequence (as determined from high-quality PacBio assemblies). The *y*-axis gives sequence counts across all four genes (AMA1, EBA175, DBLMSP and DBLMSP2) and all 14 evaluated samples. The top left panel shows the distances between the 3D7 sequence and the truth assemblies, while the other panels show the distances for sequences inferred by the evaluated tools. The dotted lines and adjacent numbers show the mean scaled edit distance

S7-10 and further confirm `gramtools` genotyping finds optimal paths in the graphs in Additional File 1: Figure S11.

### Application: unified SNP and large deletion analysis in *M. tuberculosis*

Nested variation occurs naturally when jointly genotyping small variants overlapping structural variants. We assessed how `gramtools` compares to two other genome graph tools, `GraphTyper2` [9] and `vg` [4], in one such situation in *M. tuberculosis*. We evaluated each tool's ability to genotype a fixed set of input small variants (SNPs and indels) and overlapping deletions.

We started from variant calls obtained by running `Cortex` on 1017 publicly available Illumina samples (see the "Variant discovery" section). We also produced high-quality hybrid assemblies for 17 of these samples using matched Illumina [26] and PacBio reads [27] (see the "Hybrid assembly of the 17 evaluated samples" section). These assemblies were used as ground truth for evaluating genotyping.

We identified 73 high quality large deletion calls in the 17 samples, spanning a total of 45 distinct genomic regions (confirmed using the assemblies (see the "Variant discovery" section)). We then extracted all variation in the 1017 samples overlapping these 45 regions. Together, these provide the variant sites at which we evaluate each tool.

For `gramtools`, we built a genome graph of each deletion region from multiple-sequence alignments of the `Cortex` variant calls applied to the *M. tuberculosis* H37Rv reference genome [28] (see the "`gramtools` genome graph construction" section). The graphs were then combined with the rest of the reference genome. To genotype the same variants in `GraphTyper2` and `vg`, we merged the VCF files of all 1017 input samples using `bcftools`. VCF is the required input format for genotyping in `GraphTyper2` and the only input format that worked in `vg` after failing on multiple sequence alignments (see the "`vg` and `GraphTyper2` genome graph construction" section).

We first looked for each of the 73 known deletions in each tool's VCF output and found `GraphTyper2` called all 73, `gramtools` called 70 and `vg` called 66. We then assessed each tool's ability to resolve each deletion region in the 17 evaluation samples. For each region, we applied called variants to the *M. tuberculosis* reference genome and measured edit distance to the truth assembly using `minimap2` [29] (similar results were obtained using `bowtie2` [30]; see Additional File 1: Figure S17 and the "Mapping evaluated regions to truth assemblies" section).

Figure 6 shows the cumulative distribution of scaled edit distances (edit distance divided by the length of the aligned sequence) for each tool. `gramtools` achieves the lowest mean distance to the truth (1.2%), followed by `vg`(2.4%) and `GraphTyper2` (3.2%) (as a baseline, the mean distance of the reference genome sequence to the truth is 4.8%). We note that without 8 long, false positive insertions with edit distances >0.5, the mean distance for `GraphTyper2` is 2.6%. `gramtools` also achieves the highest fraction of perfectly resolved sequences (edit distance 0) (86.7%), followed by `GraphTyper2` (69.3%) and `vg` (54.0%). A small number of sequences remained unmapped (9 for `gramtools`, 12 for `GraphTyper2` and 13 for `vg` (Additional File 1: Figure S18)).

To understand genotyping performance in more detail, we broke down called variants into different types (insertions, deletions, SNPs) and sizes and measured precision (what proportion of calls made were correct) and recall (what proportion of the expected

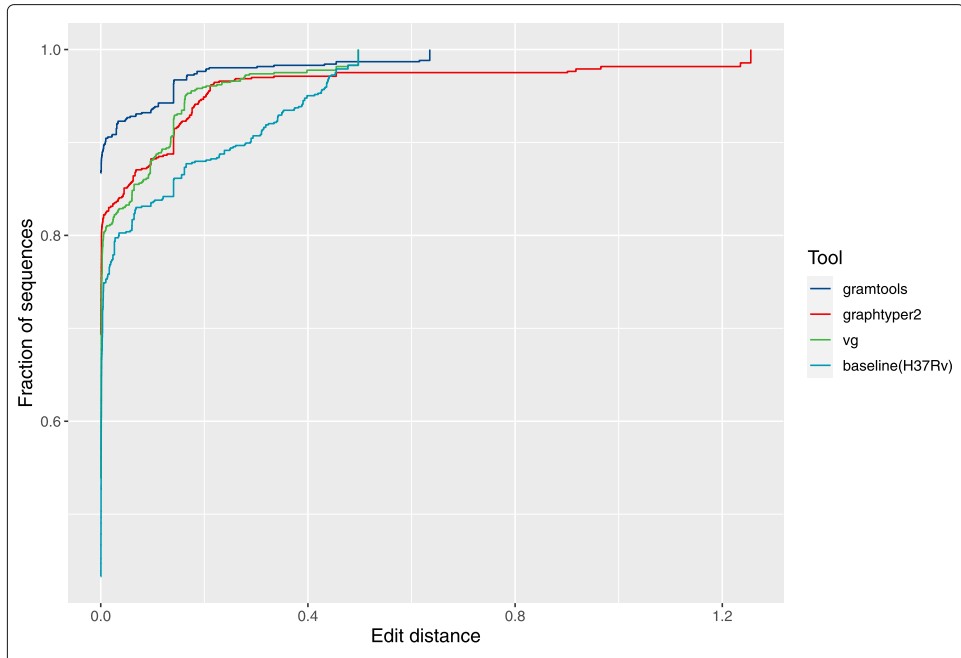

**Fig. 6** `gramtools` joint SNP and deletion genotyping performance compared to other genome graph tools. For each of the 45 deletion regions in each of the 17 validation samples, we made a sequence containing each tool's calls, giving a total of 765 data points per tool (`gramtools`, `vg`, `graphtyper`). The curves show the cumulative frequency of edit distances between these sequences and truth assemblies. Baseline refers to using the *M. tuberculosis* H37Rv standard reference sequence only

calls were recovered) (see the "Evaluating variant calls using `varifier`" section). Compared to the other tools, we found `vg` has a larger number of incorrect and missing small variants (insertions and deletions < 10bp, and SNPs). Notably, SNP recall and precision were 57.2% and 86.7% for `vg`, compared to 91.3% and 93.8% for `gramtools` and 90.0% and 99.1% for `GraphTyper2` (Additional File 1: Figure S19). Similarly, we found `GraphTyper2` has a larger number of incorrect and missed large insertions and deletion calls (> 50bp): for large deletions, `GraphTyper2` recall and precision were 67.3% and 64.4%, compared to 97.8% and 99.6% for `gramtools`, and 97.1% and 99.5% for `vg` (Additional File 1: Figure S19). `gramtools` achieved the highest recall across all variant categories but has lower precision than `vg` or `GraphTyper2` for some categories, notably

**Table 1** Computational performance of each tool

|  | Index | | | Map and genotype | |
|---|---|---|---|---|---|
|  | Disk (Mb) | Max RAM (Mb) | Speed (sec) | Max RAM (Mb) | Speed (reads/sec) |
| `vg` | 29 | 609 | 105 | 605[158] | 3,961 |
| `gramtools` | 153 | 480 | 32 | 632 | 34,290 |
| `GraphTyper2` | – | – | – | 869[88] | 7,604 |

*Index*: genome graph processing step allowing subsequent read mapping. For `vg`, includes a graph pruning step to reduce graph complexity (else temporary disk use exceeded 500 Gb before completion, see "vg and `GraphTyper2` genome graph construction" section). `GraphTyper2` has no separate indexing operation. *Map and Genotype*: Speed shows the average number of reads mapped across the 17 samples (10.7 million) divided by the average CPU time. `vg` and `GraphTyper2` have separate read mapping and genotyping steps: for speed, CPU time is summed, and for RAM, mapping is shown followed by genotyping in square brackets. `GraphTyper2` does not implement its own mapping but requires an input file of reads mapped to a linear reference genome; mapping RAM and speed is shown for `bowtie2` with default parameters. *metrics*: Mb, megabytes; sec, total CPU seconds (accounts for multi-threading, 10 threads used for genotyping in each tool)

SNPs and small (1–10bp) and mid-size (11–50bp) insertions (Additional File 1: Figure S19).

In terms of computational performance, `gramtools` processed the most reads per CPU second while using comparable amounts of RAM on this dataset (Table 1). A bottleneck in `vg` is temporary disk use, exceeding 500 Gigabytes without pruning the graph to remove densely clustered variation. For `GraphTyper2`, we include counting a separate mapping step to the reference genome (with `bowtie2`) as it is a prerequisite to genotyping (for `vg` and `gramtools`, performance includes mapping reads to genome graphs before genotyping). While `gramtools`' mapping and genotyping is 4 to 8 times faster than `vg` and `GraphTyper2` in this benchmark, we are also aware of `gramtools`' much lower mapping speed to the human genome [31]. Computational performance depends on the genome and variants under analysis and on the genome graph approach; we consider these further in the "Discussion" section.

In this experiment, `GraphTyper2` and `vg` are able to genotype variation at multiple scales. One caveat is the VCF file they genotype can contain inconsistencies. For example, if one VCF record describes a deletion and another describes an overlapping SNP, a reference call at the deletion and an alternate call at the SNP are inconsistent because the two calls imply different sequences. This occurs because the variants are related but expressed in isolation. By contrast `gramtools` models site relationships explicitly, outputting a VCF file without inconsistencies and a jVCF file mapping the nested variation.

An output format like our proposed jVCF becomes especially important when analysing more complex variation such as SNPs on top of alternate haplotypes, where variants need to be expressed against different references. We now show such an application of multiscale variation analysis using the *P. falciparum* surface antigen DBLMSP2, which would not be possible using the VCF files output by `vg` or `GraphTyper2`.

### Application: charting SNPs on top of alternate haplotypes

When two diverged forms of a gene segregate in a population, we want to access small variants on top of each. Returning to the surface antigen DBLMSP2 in *P. falciparum*—which we have shown is accurately genotyped by `gramtools` using simulated and real data—we assessed whether `gramtools`' multiscale genotyping and jVCF output could recover the two diverged forms of DBLMSP2 and access variation on top of each form. We genotyped 706 *P. falciparum* samples from Ghana, Cambodia and Laos using `gramtools` and analysed a combined jVCF file of all calls in all samples (see the "*P. falciparum* dimorphic variation analysis" section). Genotyping, including read mapping, used an average of 1.14 Gigabytes of peak RAM, processing an average of 2525 reads per CPU second.

Figure 7 shows a matrix of calls in each sample at each variant site inside the DBL domain of DBLMSP2, known to be dimorphic [32]. The tree on the left depicts a hierarchical clustering of the sample haplogroups. Its most basal split distinguishes two forms of the domain (form 1 (dark pink) and form 2 (light pink) on the right of the heatmap), with an average scaled edit distance between the two forms of 16.8% compared to within-form distances of 1.4% and 4.8%. gramtools can thus recover two divergent forms, as expected given the known dimorphism. Building a phylogenetic tree of the sequences confirmed the presence of the two forms and showed high concordance with the clustering tree in how samples are assigned to each form (Additional File 1: Figure S20).

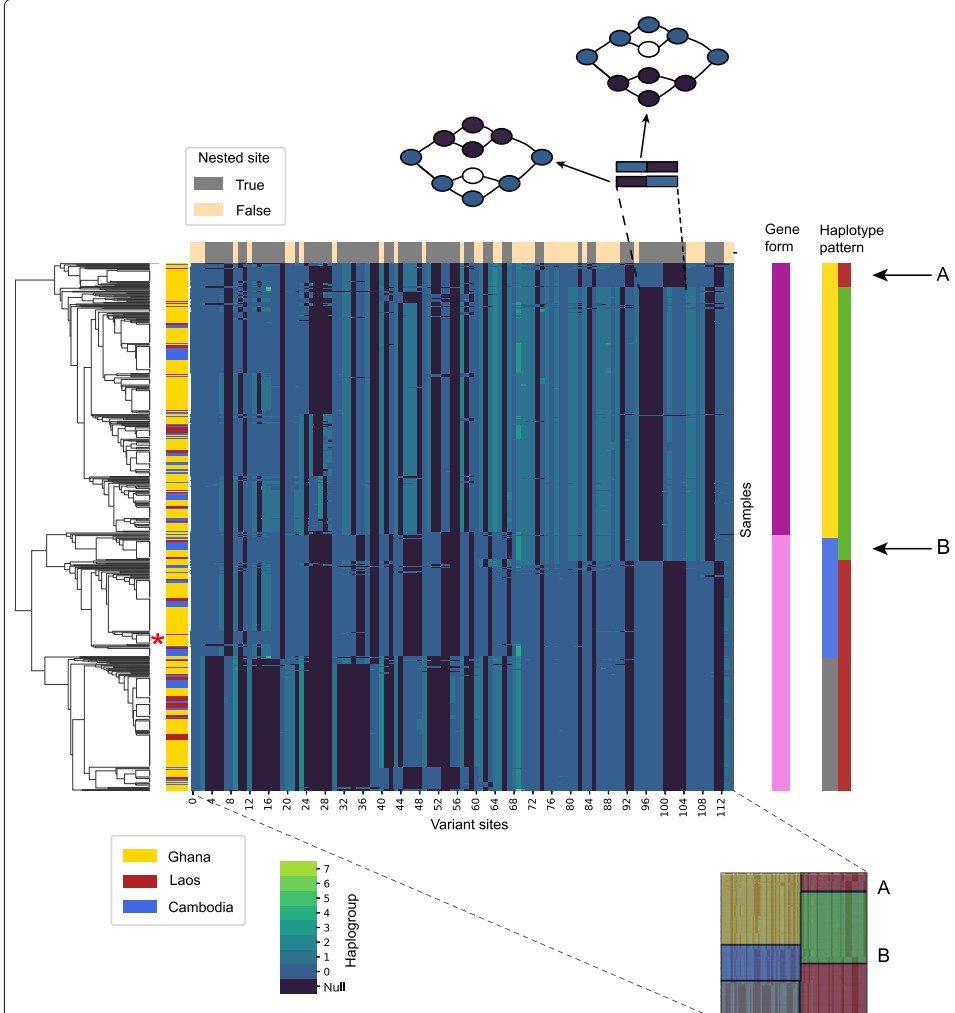

**Fig. 7** `gramtools` captures allelic dimorphism and nested variation in *P. falciparum* gene DBLMSP2. Genotypes of 706 samples from Ghana, Laos and Cambodia spanning the DBL domain of gene DBLMSP2 are represented in a heatmap of variant sites (*x*-axis) versus samples (*y*-axis). Each cell in the main square is coloured by haplogroup, which can be considered here as an alternate allele number; a null (darkest blue) haplogroup indicates no call made at a site. The tree on the left shows a hierarchical clustering of alleles. The clade nearest to the reference genome 3D7 is shown with a red asterisk to the right of the tree. The vertical strip to the left of the heatmap shows country of origin of each sample. The vertical strip to the right of the heatmap shows the broad classification of haplotypes into two forms—form 1 (dark pink) and form 2 (light pink), corresponding to the deepest split in the tree and the two known dimorphic forms. Comparing the location of the asterisk shows 3D7 is of form 2. In order to linearise the sites within the graph for display in a heatmap, they are sorted topologically, and a strip across the top of the heatmap shows whether sites are nested. As a clarifying example, the blowout at the top shows how two incompatible nested SNPs on different backgrounds are displayed as either dark-then-light-blue or light-then-dark blue blocks in the heatmap, with the incompatible haplotype shaded darkest blue (null call). This heatmap allows visualisation of haplotype patterns and recombination. We show, bottom right, a blowout with 5 haplotype patterns coloured yellow, blue, grey (all from the left side of the heatmap) and green, red (from the right). Using these classifications, we describe all alleles as combinations of two haplotype patterns, shown in the far-right vertical strip. Form 1 is almost entirely yellow-green, and form two itself has two subforms—blue-red and grey-red. We highlight two recombinant haplotype patterns labelled A (yellow-red) and B (blue-green). Both A and B exist in all 3 countries

`gramtools` also produces calls in nested sites, shown as dark bars at the top of the matrix. As an example, we provide an illustrative blowout at the top of the heatmap of a region where variants are nested and occur on different sequence backgrounds. The graphs illustrate how the nested sites are called mutually exclusively: when one of the gene forms has a genotype call, the other form receives a null call (shown as the darkest blue-coloured cells), and vice-versa. This is essentially showing that there are SNPs on both genetic backgrounds (the dimorphic types) and `gramtools` is genotyping variants irrespective of background.

Interestingly, the heatmap in Fig. 7 also shows, for the first time to our knowledge, clear evidence of recombination between the two forms of DBLMSP2. At the bottom-right of the figure, we show a blowout of the matrix that is coloured by broad haplotype patterns, with form 1 being predominantly yellow followed by green and form 2 being predominantly blue or grey followed by red. This reveals two sets of samples (labelled A,B by the blowout) which are inter-form recombinants. Those labelled A have a yellow haplotype (form 1) followed by red (form 2) and those labelled B have blue (form 2) followed by green (form 1).The leftmost column of the matrix is coloured by the country of origin of each sample. Strikingly, for each group of samples (i.e. A and B) almost identical recombinants exist across each of Ghana, Laos and Cambodia. We do not yet know if these inter-form recombinants derive from a single recombination event (either ancestral to the samples or transmitted across countries) or if this recombination event has occurred multiple times independently.

Finally, we indicate, using a red asterisk on the right of the tree, the closest clade to the reference genome sequence 3D7. Leveraging `gramtools`' graph model and jVCF output, we can move beyond a 3D7-only reference-based analysis that would neither genotype nested variants on the two different backgrounds nor reveal the inter-form recombinants.

## Discussion

Genetic variation occurs through different mechanisms at different scales ranging from SNPs to large structural variants. The need to jointly analyse SNPs and structural variants therefore arises immediately when trying to genotype a cohort. We have presented a method for identifying, calling and outputting such variation in `gramtools`. By identifying site relationships in the graph, `gramtools` is able to genotype incompatible sites mutually exclusively and to output variation both against the standard reference genome and against locally defined alternate references.

One of the challenges of extending linear references with graphs is recognising modelling assumptions. Working from a single reference implicitly assumes that individuals within a species have genomes that are close to the reference. When addressing this model's limitations by moving to a graph, we are forced to make new modelling choices. At one end of the spectrum is the `gramtools` approach: genome graphs must be nested, directed acyclic graphs. This simple model allows direct access to two key notions we want to use: easily distinguishing horizontal (paralog-like) and vertical (ortholog-like) variation and defining distinct alternate sequence backgrounds. At the other extremes are very general sequence graphs with no ordering, whether De Bruijn (which collapse all repeats of size k) or `vg`-like (bidirected and allowing cycles). These models better support

more complex events such as duplications and inversions, with an added cost in complexity of implementation (e.g. reconstructing variant sites, identifying coordinates and mapping reads).

Choice of graph model also affects the computational performance of read mapping. Data structures supporting linear-time exact matching exist for a restricted class of graphs (see Wheeler graphs [33], which include De Bruijn graphs). In gramtools, graphs do not have this property, and we use a data structure (the vBWT) with run time and memory use that can scale exponentially with genome size and density of stored variation. In practice, however, gramtools shows good computational performance on the *P. falciparum* and *M. tuberculosis* graphs used in this study. Here, we analyse variation from a few thousand samples and a small number of regions, but gramtools has also successfully been used to genotype 70,000 (whole genome) *M. tuberculosis* samples in a graph containing 1.25 million variants (one variant every 3.5 bases) [34]. We are also, however, aware of gramtools' very low mapping speed on the human genome [31]. More work is required in the pangenomics field to understand the different real-world performance challenges of repetitiveness (*P. falciparum* is much more repetitive than human), genome size (microbes are tiny but diverse) and amount, type and density of variation.

Applying gramtools' genome graph model on microbial datasets, we obtain three main results. First, in *P. falciparum* genes with high diversity, gramtools genotyping with genome graphs outperforms reference-genome-based callers. Second, gramtools provides superior genotyping accuracy compared to genome graph tools vg and GraphTyper2 when jointly genotyping large deletions and overlapping small variants in *M. tuberculosis*. (We note that during the finalisation of this paper, a new caller based on vg named Giraffe [8] was released, which we have not tested here). Third, we show how locally defined alternate references allow accessing small variants on top of diverged forms of a dimorphic gene in *P. falciparum*.

These results highlight three central concepts for genome graph based analyses: compatibility, consistency and interpretability.

First, while genome graphs extend beyond a single linear reference, maintaining compatibility with linear references is essential. gramtools outputs variation in terms of the standard reference genome in a VCF file and also produces a personalised reference genome, allowing reference-based callers to discover previously inaccessible variation. Many genomic analyses rely on a linear reference, which provides a simple coordinate system for referring to genomic annotations and comparing individuals. Recently, the rGFA format for describing genome graphs was proposed [35]; starting from a central linear reference, it assigns stable names and offset coordinates to alternate references. rGFA is a valuable and complementary idea to the jVCF described here: it assigns coordinates and references in constructed genome graphs, while our jVCF describes sites and called variation in genotyped genome graphs. Like the graphs used by gramtools, rGFA works on globally linear graphs in order to maintain clear homology relationships. Although this feature is not implemented, jVCF could easily be extended to store rGFA-defined alternate references, allowing for expressing variant calls against any reference.

Second, genome graphs offer the opportunity to genotype cohorts of samples consistently. By representing all variation found in a set of samples, they can be used to produce a full sample-by-site matrix. gramtools achieves this by detecting all variant sites in

the graph and outputting them, along with their relationships, in a jVCF call format. Previous work has explored graph decomposition into a fixed set of variant sites [36] and is available in vg with the *deconstruct* command. However, vg genotyping currently does not output all such sites nor define and output alternate references. gramtools provides (to our knowledge) the first working implementation of consistent graph to variant site mapping.

An important determinant of compatibility and consistency is the graph construction process. In gramtools, we use our tool make_prg [15]. From a multiple sequence alignment, make_prg collapses common sequence between samples, clusters the remaining sequence into subgroups, and repeats the process recursively. This algorithm provides two main advantages. First, it limits recombination to similar input haplotypes, which reduces combinatorial explosions in variant dense regions, a source of computational bottlenecks and graph ambiguity [37]. Second, it naturally creates a hierarchy between sites as they are gradually defined on different sequence backgrounds. This captures incompatibility between sites (as in SNPs under a large deletion) as well as the process of divergent sequence evolution.

Finally, while single references and VCF provide good interpretability, we show how analysing two diverged forms of a dimorphic surface antigen in *P. falciparum* (DBLMSP2) benefits from locally defined alternate references. In contrast to existing sequences such as the alternate MHC loci in the human reference genome [38], here these are tied together in a graph-based framework. Outputting variation on different sequence backgrounds can provide finer resolution than with a single reference and will enable studying the functional impact and population genetics of nested variants.

## Conclusions

We provide a framework for identifying and genotyping multiscale variation in genome graphs and show its successful implementation in gramtools. We find good genotyping performance compared to state-of-the-art genome graph tools GraphTyper2 and vg and additionally provide an analysis of allelic dimorphism using multiple references which to our knowledge can only be performed by gramtools.

Multiscale variation analysis goes hand in hand with the gradual extension of reference genomes beyond their linear coordinates. Accessing this complex variation requires careful genome graph construction and stable names and coordinates for referring to alternate references. It also calls for new developments in variant call output formats, a proposal of which we implement and use in gramtools.

## Methods

### Graph definitions

Here, we formally define a variant site and the type of graph that gramtools can support. Let $G = (V, E)$ be a directed acyclic graph (DAG) with a unique minimal and unique maximal element, i.e. $G$ has a unique source and unique sink. Each node $v$ has a number of ingoing edges $\deg^-(v)$ and a number of outgoing edges $\deg^+(v)$. Define a node $v$ to be opening if $\deg^+(v) > 1$ and closing if $\deg^-(v) > 1$. Note that a node can be both opening and closing.

Let $s$ be the sink of $G$. Given any opening node $v$, let $S$ be the set of nodes that are in every path from $v$ to $s$, excluding $v$ itself. Then, $S$ is non-empty because $s$ belongs to $S$. Let $a$ and

*b* be any elements of *S*. Then, by definition of *S*, there exists a path that contains both *a* and *b*. Therefore, using the partial order defined by the edges of *G*, *a* and *b* are comparable and it follows that *S* is a totally ordered finite non-empty set. Therefore, *S* contains a unique minimal element, which we denote $c(v)$. Informally, $c(v)$ can be thought of as the first node that "closes" all paths from *v*. Similarly, given a closing node *u*, we define $o(u)$ to be $c(u)$ applied to the transpose of *G*. Informally, $o(u)$ is the node that "opens" *u*. See Additional File 1: Figure S2 for an example of how *S* and closing nodes are identified.

We use the notion of opening and closing nodes to define a variant site as follows.

**Definition.** Let *G* be a DAG with a unique source and unique sink. A *variant site* is defined as the subgraph of *G* induced from $\{u, c(u)\}$ or from $\{o(v), v\}$, where *u* is any opening node and *v* is any closing node of *G*.

We remind the reader that for any DAG *G*, there exists at least one ordering of all the nodes $v_0, v_1, \ldots, v_n$ such that given any edge $(v_i, v_j)$ of *G*, $v_i$ appears before $v_j$ in the ordering. This is called a *topological ordering* of *G*. Using the above definitions, we can now define the type of graph that is supported by gramtools, which we call a "nested directed acyclic graph".

**Definition.** Let *G* be a DAG with a unique source and unique sink. *G* is said to be a *nested directed acyclic graph* (NDAG) if there exists a topological ordering of all nodes $v_0, v_1, \ldots, v_n$ such that adding brackets to this ordered list of nodes according to the following rules results in balanced opening and closing brackets:

1   For each opening node *u*, add $[_u$ after *u*, and add $]_u$ before $c(u)$;
2   For each closing node *v*, add $[_v$ after $o(v)$ and add $]_v$ before *v*, unless these brackets were already added by case 1.

Note that each matching pair of brackets in the above definition corresponds to one variant site. See Additional File 1: Figure S3 for an illustration.

To be able to index with the vBWT, `gramtools` would apply the following modifications to the graph, producing a new graph where there is a one-to-one correspondence between the set of opening and closing nodes. Specifically, this means that a node is either opening or closing and cannot open or close more than one node. Essentially, the method entails adding a new node to the graph for each balanced bracket that was added to the topological ordering of the nodes. Starting from the innermost brackets, for each matching pair of brackets $[_a$ and $]_a$, where node *a* precedes $[_a$ and node *b* follows $]_a$ in the topological ordering with balanced brackets (so we are considering $\ldots, a, [_a, \ldots, ]_a, b, \ldots$):

- Add a node called $[_a$ with no sequence and an edge $(a, [_a)$ to the graph and move the outgoing edges of *a* to $[_a$;
- Add a node called $]_a$ with no sequence and an edge $(]_a, b)$ to the graph and move the incoming edges of *b* to $]_a$.

See Additional File 1: Figure S4 for an illustration of this process.

In practice, `gramtools` does not need to transform an NDAG or verify if an input DAG is an NDAG, as it takes as input constructed graphs that are already indexable NDAGs. This is achieved using one of two ways described below.

### Genome graph construction and `make_prg`

`gramtools` can construct genome graphs without nested variation from a reference genome and a VCF of variants. Overlapping records in the VCF file are merged by enumerating all possible combinations up to a specifiable limit. This method creates an NDAG because no variant sites overlap, giving a natural balanced bracket representation of sites. However, this approach rapidly fails in variant-dense regions or for large cohorts of samples due to a prohibitively large number of allele combinations. We solve this problem by allowing for nested variation. To build nested graphs, we apply an algorithm called recursive collapse and cluster (RCC) starting from a multiple-sequence alignment. RCC was first introduced in the context of bacterial pan-genomic tool `pandora` [15] and is implemented in and available at https://github.com/iqbal-lab-org/make_prg.

RCC identifies invariant regions of a given minimum size and collapses them into a single graph node. The remaining regions form variant sites, and each gets clustered based on their *k*-mer content. This procedure is repeated recursively on each cluster, until either a maximum nesting level is reached or the sequences are too small (in which case they are directly enumerated as alternative alleles). In this way, variants appear in subsets of samples with similar sequence backgrounds. The RCC algorithm generates hierarchically nested sites by construction: each cluster of sequences corresponds to one variant site, the clustering process generates distinct clusters, and recursive sequence collapsing occurs fully inside of a cluster, making new clusters nested. RCC thus produces an output graph that is an NDAG. We provide an illustration of RCC and how it induces the balanced bracket representation of an NDAG in Additional File 1: Figure S5.

Two command-line parameters affect what graph gets produced. First, *max_nesting* is the maximum number of collapse and cluster recursions to perform, which gives the maximum number of nesting levels in the graph. Second, *min_match_length* is the number of invariant bases between samples for them to be collapsed in a single node. Sequence collapse is what allows paths coming before and after to cross; a larger value thus reduces recombination between the input haplotypes. This provides a way to control combinatorial path explosions in the graph.

### vBWT data structure in `gramtools`

The vBWT data structure marks variant sites with numeric identifiers so that alleles get sorted and queried together in the suffix array (Fig. 8a). This representation induces branching at each site entry and exit such that mapping has worst-case exponential runtime. To speed mapping, we seed reads from an index storing the mapped intervals of all sequences of a given size *k*. Linear-time exact match indexes on genome graphs exist (e.g. GCSA [39]) but require a prefix-sorting step that is worst-case exponential.

vBWT's numeric identifiers are also used for recording mapped read coverage along variant sites (Fig. 8b). Coverage recording handles two types of uncertainty: horizontal, where sequence is repeated across the genome, and vertical, where sequence is repeated in alleles of a site. To handle horizontal uncertainty, we randomly select one read mapping instance, as is typically done in standard aligners [30]. To handle vertical uncertainty we store allele-level equivalence class counts which are counts of reads compatible with groups of alleles, an idea introduced in `kallisto` [16]. This allows allelic uncertainty to be accounted for during genotyping. Per-base coverage is also stored on the graph (Fig. 8) and used during genotyping.

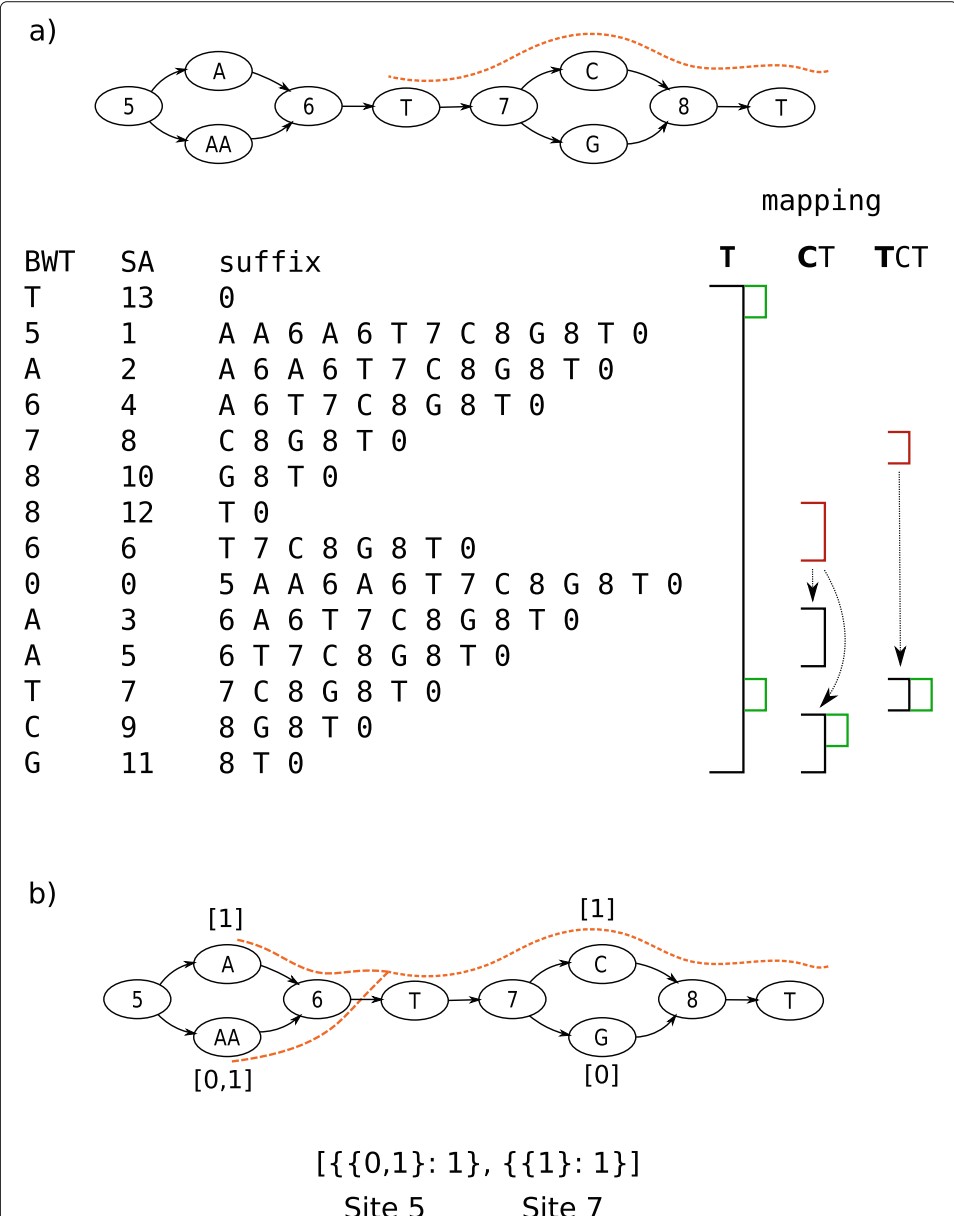

**Fig. 8** `gramtools` mapping and coverage recording. **a** Variant-aware Burrows Wheeler Transform (vBWT). Each row of the text matrix encodes one position in a linear representation of the graph. BWT: stores the character in the previous position; SA: suffix array, stores the position in the text; suffix: stores the text from SA position to the end. Two markers are used for every variant site in the genome graph: odd markers mark site entry and even markers allow alleles to sort and be queried together. Black intervals mark regular BWT backward searching, with each match to the currently mapped base shown in green. Arrows from red intervals mark vBWT-specific jumps in and out of sites, making the search branch. The read being mapped is shown in dashed orange. **b** Square brackets under allele nodes show per-base coverage storage. Another array shown below stores allele-level coverage at each site. Mapped reads increment equivalence class counts representing compatibility: in this example, the read is compatible with both alleles 0 and 1 at site 5 and only with allele 1 at site 7. Both kinds of coverage are used in genotyping

### Genotyping model

The genotyping model in `gramtools` supports haploid and diploid genotyping. It assigns a likelihood to each candidate allele (or pair of alleles for diploid) computed from base-level and allele-level read coverage.

`gramtools` stores coverage in equivalence classes, following ideas from [16]. Let $A$ be the set of alleles at a variant site. We partition the set of all reads overlapping $A$ into subsets, i.e. equivalence classes, where all reads belonging to one subset map perfectly to the same subset $X$ of $A$ (e.g. reads that map uniquely to allele 1, or reads that map equally well to alleles 1 and 2; see Fig. 8b). For each equivalence class, we store a count $c_X \in \mathbb{N}$ of reads compatible with the alleles in set $X$, and for each mapped read, we increment its $c_X$ at each overlapped site. If a read has multiple (horizontal) mapping instances, we select one at random, and the counts $c_X$ are incremented as above. When a read's count $c_X$ is incremented, for each allele $a \in X$, the count of each base the read mapped to is also incremented. Base-level counts are written $c(a_i)$, where $a_i$ is the $i$th base of allele $a$.

Coverage compatible with allele $a$ of length $l_a$ is defined as $c(a) = \frac{1}{l_a} \sum_{i=1}^{l_a} c(a_i)$ and incompatible coverage as $i(a) = \sum_{X \subset A : a \notin X} c_X$. In this way for each candidate allele, we capture a per-base correct coverage generation process as well as an incorrect coverage generation process on incompatible alleles.

We model the expected per-base read coverage in a site using an estimate of the mean $\lambda$ and the variance $\sigma^2$ of true coverage across all variant sites. For each site, true coverage is estimated as the average per-base coverage of the allele with the most coverage. If $\sigma^2 \leqslant \lambda$, we model observed coverage as coming from a Poisson distribution:

$$p(c(a) = k | \lambda) = e^{-\lambda} \frac{\lambda^k}{k!}$$

Else, we use the negative binomial (NB) distribution

$$p(c(a) = k | p, r) = \binom{k + r - 1}{k} (1 - p)^r p^k$$

When using the NB distribution, we need to estimate the standard parameters of the NB distribution, $r$ and $p$. $r$ is estimated from rearranging the formula for the expected variance of NB as $r = \frac{\lambda^2}{\sigma^2 - \lambda}$, and similarly $p$ is estimated from the expected mean of NB as $p = \frac{\lambda}{\lambda + r}$.

We model incorrect coverage $i(a)$ as coming from sequencing errors with rate $\epsilon$: $p(i(a) = k | \epsilon) = \epsilon^k$. $\epsilon$ is estimated from the mean base quality score in the first 10,000 processed reads.

We also use per-base coverage to penalise gaps in coverage. Given a function $g(a)$ returning the number of zero-coverage positions in allele $a$, the probability $p(g(a) = k)$ of seeing $k$ gaps if $a$ is the true allele is $p(zero\_cov)^k$, where $p(zero\_cov)$ is obtained by computing $p(c(a))$ using the above formula with $c(a)$ set to zero. In practice, we use $\frac{k}{l_a}$ as the exponent instead of $k$ so as not to penalise long alleles.

These three terms combined give the likelihood of allele $a$

$$\mathcal{L}(a) = p(c(a))p(i(a))p(g(a))$$

The allele $a'$ that gets called is the maximum-likelihood allele, and we define the genotype confidence of the call as

$$\min_{a \in A : a \neq a'} \frac{\mathcal{L}(a')}{\mathcal{L}(a)}$$

which is the likelihood ratio of the called allele and the next most likely allele.

This holds for haploid genotyping. For higher ploidy, the likelihood function generalises to a set of alleles $S$ as

$$\mathcal{L}(S) = p(i(S)) \prod_{a \in S} p(c(a)) p(g(a))$$

where $i(S) = \sum_{X \subset A : X \cap S = \emptyset} c_X$.

For diploid genotyping, $S = \{a_1, a_2\}$ and $p(c(a))$ is parameterised by $\frac{\lambda}{2}$ because we expect half the total site coverage on each of two called alleles.

### Nested genotyping

The `gramtools` genotyping model is applied recursively from child sites to their parent sites. Calls in child sites restrict the set of alleles considered in the parent so that the number of choices is reduced: for each outgoing path from a site, $\prod_i^n p_i$ paths are considered, where $p_i$ is the number of distinct alleles called at site $i$ (e.g. in diploids 0, 1 or 2) and $n$ the number of child sites encountered. Some extra paths are also retained when genotype confidence for a child site is low, in order to propagate uncertainty to parent calls. If there are more than 10,000 possible alleles, only the 10,000 most likely alleles are considered. This does not require enumerating all possible alleles as the most likely alleles in child sites have already been computed.

An example of the nested genotyping procedure is shown in Fig. 9. To maintain coherence, if child sites on two different branches of a parent site are genotyped, whole branches can get invalidated. For example at a ploidy of one if an outgoing branch from a parent site is called, all children sites on the other branches receive null calls.

### *P. falciparum* surface antigen graphs and genotyping validation
#### *Graph construction*

We started from VCF files produced by running `Cortex`, a de novo assembly-based variant caller [5], on read sets of 2,498 samples from the Pf3k project [20] (all reads are publicly available on the ENA, see the Availability of data and materials section). `Cortex` has a very low false positive call rate and can call the divergent forms of *P. falciparum* surface antigens [22]. The `Cortex` independent workflow was run, using the bubble caller with *k=31*. The VCF files are publicly released on zenodo (see Availability of data and materials) and `Cortex` is publicly available at https://github.com/iqbal-lab/cortex.

For each surface antigen gene (DBLMSP, DBLMSP2, EBA175 and AMA1), we generated sequences for each sample by applying `Cortex` variants at the gene coordinates, plus 5000 bp on each side, to the *P. falciparum* 3D7 reference genome. We generated multiple-sequence alignments of each gene using `mafft` [40] and passed them as input to our construction tool `make_prg`. For the simulation experiment, two graphs were built for DBLMSP and DBLMSP2 with maximum nesting levels of 1 and 5. The graphs without nesting have 451 and 413 variant sites for DBLMSP and DBLMSP2, and the graphs with nesting have 558 and 500 variant sites respectively.

*Path and read simulation*

From each non-nested graph, 10 paths were simulated and threaded through the nested graph using `gramtools`' `simulate` command. This results in jVCF files recording the true genotypes for each path in each graph. Illumina HiSeq25 75-bp reads (0.023% per-base error rate) were simulated from each unique path using ART [41] at 40-fold coverage, genotyped using `gramtools` and calls evaluated by comparing the genotyped and truth jVCF files.

The nested graph contains more paths than the non-nested graphs due to allowing greater recombination between variant sites. We therefore simulated paths from the non-nested graph to ensure each path exists in both graphs.

*Comparison with reference-based callers*

The nested graphs of the four surface antigens DBLMSP, DBLMSP2, EBA175 and AMA1 were combined in positional order with the rest of the reference genome, using a custom script (see the Availability of data and materials section).

For each of the 14 validation samples, we ran `SAMtools` and `Cortex` using the *P. falciparum* 3D7 reference genome and `gramtools` using the surface antigen genome graph. For each genotyped sample, `gramtools` infers a haploid personalised genome

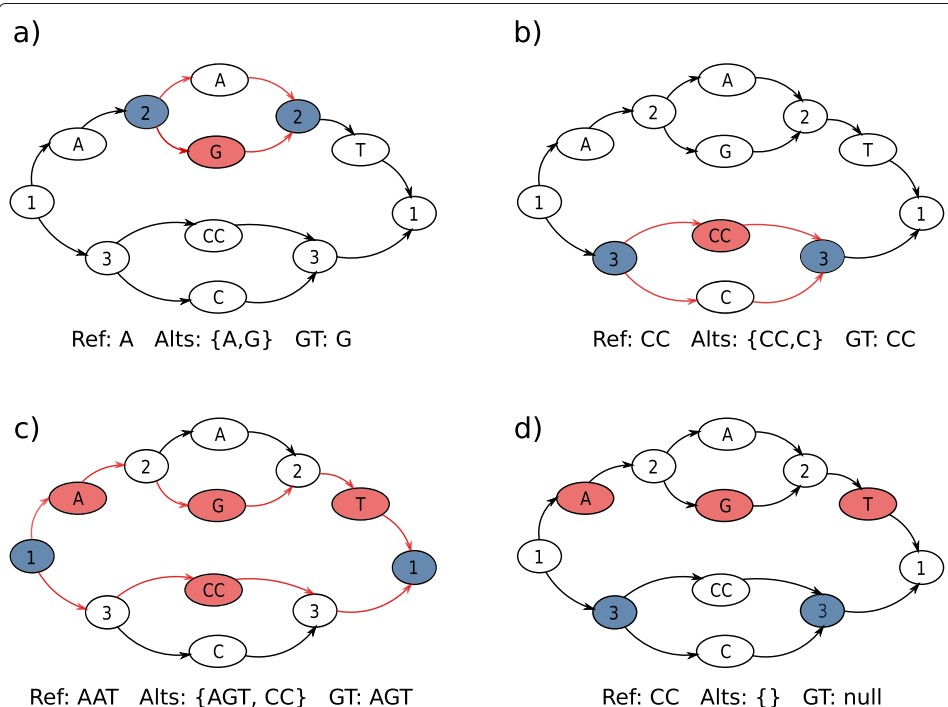

**Fig. 9** Nested genotyping procedure. Nodes with numbers mark the start and end of variant sites. In each panel, blue-filled nodes mark which site is being processed, red-filled nodes mark called alleles, and red paths mark alleles considered for genotyping. *Ref* is the reference allele, *Alts* are the alleles considered for genotyping, and *GT* is the called genotype. The example shows haploid genotyping. **a** Genotyping of child site 2. Allele "G" gets called. **b** Genotyping of child site 3. Allele "CC" gets called. **c** Genotyping of parent site 1. *Alts* are generated from the alleles called in child sites 2 and 3: allele "G" is used from site 2, and allele "CC" is used from site 3, producing alleles "AGT" and "CC". Allele "AGT" gets called, going through site 2. **d** In c, the path going through site 2 was called. Because genotyping is haploid, the call at site 3 is invalidated (GT becomes "null")

(PR) as the whole-genome path taking the called allele at each variant site. `SAMtools` and `Cortex` were then run once more using the personalised reference instead of 3D7.

When mapping the gene sequence with variants applied to the truth assemblies, we measure performance as the edit distance reported by `bowtie2` (version 2.4.1) divided by gene length.

### *M. tuberculosis* SNP and large deletion analysis
### *Hybrid assembly of the 17 evaluated samples*

Each sample was initially assembled using `Unicycler` [42] and `Canu` [43], followed by `Circlator` [44] using the corrected reads output by `Canu`. `Unicycler` version 0.4.8 was used with the option '`-mode bold`', Illumina reads given using the options '`-short1` and '`-short2`', and the PacBio subreads using the '`-long`' option. `Canu` version 2.0 was used with the option '`genomeSize=4.4m`' and the PacBio reads provided with the option '`-pacbio-raw`'. The only exception was sample N1177, which was initially assembled using `Flye` [45] version 2.8-b1674 with the PacBio subreads input with the option '`-pacbio-raw`'.

The initial assembly for each sample was chosen for further manual polishing based on inspection of mapped reads and comparison with the H37Rv reference genome. The `Unicycler` assembly was used for samples N0004, N0091, N0155, N0157, N1283, N0072, N1202 and N0153. The Canu assembly was used for samples N0031, N1176, N0052, N0136, N0146, N1216, N1272 and N0054. Redundant and/or contamination contigs were removed from samples N0072, N1202, N0052, N0136, N0146, N1216 and N1272. Manual fixes were applied to samples N0054 and N0153 by breaking contigs at errors, with the aid of the Artemis Comparison Tool (`ACT`) [46], and re-merging using `Circlator` using the default settings. Next, `Pilon` [47] (version 1.23) was run iteratively on each assembly using the Illumina reads as input, mapped with `BWA-MEM` [48] (version 0.7.17-r1188, default settings) until no more corrections were made, up to a maximum of 10 iterations. Finally, the 'fixstart' function of `Circlator` was used to ensure that each assembly began with the *dnaA* gene, for consistency with the H37Rv reference genome. The result was all 17 samples assembled into a single, circularised contig.

### *Variant discovery*

We first obtained variant calls from the read files of the 17 evaluated samples and an additional 1000 samples available in the ENA (see the Availability of data and materials section). We ran `Cortex` to obtain the calls, using our lab's wrapper `clockwork` version 0.8.3, publicly available at [https://github.com/iqbal-lab-org/clockwork](https://github.com/iqbal-lab-org/clockwork). `clockwork` runs `Cortex`'s independent workflow using the bubble caller with *k=31*. The VCF files are publicly released on zenodo (see Availability of data and materials).

`Cortex` identified a total of 73 deletions in the 17 evaluated samples, between 100 and 13,000 bases in length and falling in 45 distinct genomic regions. To validate the calls, we mapped their corresponding long-read assemblies to the *M. tuberculosis* H37Rv reference genome with `minimap2`, which validated 68. The remaining 5 were manually confirmed using `ACT`: for each sample we mapped the short reads to the reference genome and to the assembly using `bowtie2` and mapped the assembly to the reference using `nucmer` [49]. In `ACT`, we view all three together and validate a deletion when it appears in the

assembly-reference mapping at the expected coordinates and when read pileups confirm the event. These are shown in Additional File 1: Figure S12-16.

Having validated all the deletions, we extracted all `Cortex` calls occurring under the 45 deletion regions in the 1017 samples, giving us a joint set of large deletions and overlapping SNPs and indels.

### `gramtools` *genome graph construction*
We built one genome graph for each of the 45 regions identified as containing large deletions in our 17 evaluation samples. As for the *P. falciparum* surface antigen graphs, for each region, we applied `Cortex` calls to the *M. tuberculosis* H37Rv reference genome, generated multiple sequence alignments with `mafft`, passed them as input to our construction tool `make_prg` and combined them with the rest of the genome.

### `vg` *and* `GraphTyper2` *genome graph construction*
We set on building a `vg` genome graph from the same multiple sequence alignments (MSA) used by `gramtools` to maximise comparability. Using `vg` version 1.26.0, we built each of the 45 regions from MSA using `vg construct` and combined them with the invariant parts of the *M. tuberculosis* H37Rv reference genome using `vg concat`.

Indexing this graph, a prerequisite to read mapping and variant calling, used >10 Terabytes of temporary disk space before we stopped it. We deemed > 500 Gigabytes prohibitive and set that as a limit. We ran `vg prune` to remove densely clustered variation from the graph and, after exceeding 1 Terabyte of disk indexing the pruned graph using default parameters, successfully indexed the pruned graph with parameters `-k10 -X3`.

We then ran `vg call` for each of our 1017 samples against the MSA graph. However, after successful mapping to this indexed graph, `vg call` failed with a segmentation fault.

We therefore built a graph from a VCF file instead. We ran `vg deconstruct -p -e` to obtain a VCF file describing the variants identified by `vg` in the `vg` MSA-constructed graph, and manually validated the variation using one sample when compared to `gramtools`. However, running `vg construct` with this VCF also failed with a segmentation fault.

We therefore used `vg` graph construction and genotyping from a merged VCF of all variants in the 45 regions which we produced using `bcftools`. This ran successfully after graph pruning to stay under our disk limit.

This VCF file was also used as input to `GraphTyper2` via `graphtyper` version 2.5.1, running its `genotype_sv` subcommand. `GraphTyper2` only accepts VCF files as input and not MSAs.

### *Covered positions and number of variants*
Altogether, the 45 deletion regions cover 51,701 bp of the reference genome. The variants under them cover 4105 reference positions in 1,109 sites in the `gramtools` graph and 2,386 positions in 1434 sites in the merged VCF file used by `vg` and `GraphTyper2`.

### *Mapping evaluated regions to truth assemblies*
We evaluated a total of 3,060 sequences by mapping them to truth assemblies: 17 samples x 45 regions x 4 tools (`gramtools`, `vg`, `GraphTyper2` and the reference genome sequence). Using `bowtie2`, 10.4% all sequences failed to be fully aligned due to excessive divergence between the called sequence and the truth.

To recover more alignments, we used `minimap2` which is designed to align more highly diverged sequences (such as ONT long reads) [29]. For each evaluated sequence, we took the alignment with the greatest number of matches to the assembly and extracted assembly sequence of the same length from the first aligned position (including soft- or hard-clipped). We obtained the edit distance between the two sequences from Needleman-Wunsch alignment using `edlib` [50]. Using this approach reduced the proportion of unaligned sequences to 1.1%.

To ensure evaluated alignments are unambiguous, we filter them by MAPQ $\geq 30$ so that the probability they are non-unique is $\leq 10^{-3}$ as estimated by `minimap2`. This removed 0.62% of the evaluated sequences. For each tool, 13 of 765 sequences were not mapped or had insufficiently high mapping quality. The number of unmapped and low MAPQ sequences for each tool are shown in Additional File 1: Figure S18.

We required the VCF records output by each tool to have a FILTER status set to "PASS". This changed results only marginally, giving the same number of unmapped and low MAPQ sequences and a decreased mean edit distance by 0.11% for `GraphTyper2`, 0.08% for `gramtools` and no differences for `vg`.

### *Evaluating variant calls using* `varifier`

`varifier` is a tool for measuring accuracy of variant calls in a VCF using a reference genome and a truth assembly. Given a variant call, `varifier` determines if it is correct by aligning the reference genome sequence with called variant applied (plus some flanking sequence to make the alignment specific) to the truth assembly. To compute call precision, which we define as the fraction of calls made that are correct, this procedure is applied to each variant in the evaluated VCF file. To compute call recall, which we define as the fraction of calls made out of all expected calls, `varifier` first aligns the reference genome to the truth assembly to derive a set of expected variant calls. Each expected call is then evaluated (by the mapping procedure above) against an 'induced truth genome' obtained by applying the evaluated VCF call's variants to the reference genome sequence. Then, if the expected call has been made in the evaluated VCF, it will be found in the induced truth genome. For both recall and precision, we restricted the evaluation to calls with FILTER set to "." or "PASS" in order to ignore low-confidence calls. This led to an improvement in recall and precision of 0% and 1.9% (`gramtools`), 0.1% and 2.1% (`graphtyper2`) and 0.9% and 1.1% (`vg`) on average across the variant types.

`varifier` is described in more detail in [51] and available at https://github.com/iqbal-lab-org/varifier.

### *P. falciparum* **dimorphic variation analysis**

We took a subset of 706 samples from Ghana, Cambodia and Laos out of the 2498 samples used to build the *P. falciparum* genome graphs (see the "Graph construction" section). Using `gramtools`, each sample was genotyped using the graph containing four surface antigens (DBLMSP, DBLMSP2, EBA175, AMA1). We combined each sample's jVCF output file into a single multi-sample jVCF file using `gramtools`' `combine_jvcf` executable and analysed the final jVCF using custom scripts (see the Availability of data and materials section).

## Supplementary Information

---

**Additional file 1:** Contains all supplementary text and figures for this paper.

**Additional file 2:** Review history

---

### Acknowledgements

The authors thank Rachel Colquhoun for the algorithms and first development of `make_prg`, Sorina Maciuca for the vBWT data structure and algorithms behind `gramtools` and Robyn Ffrancon for software engineering in `gramtools`. We would also like to thank the anonymous reviewers, whose detailed feedback led to considerable improvements in the manuscript.

### Peer review information

### Review history

The review history is available as additional file 2.

### Authors' contributions

BL developed `gramtools`, performed the analyses, drafted the manuscript and wrote the repository for the study (see the Availability of data and materials section). MH produced the *M. tuberculosis* assemblies and `Cortex` VCF files, helped using `ACT` and developed `varifier`. ZI designed the study and produced the *P. falciparum* `Cortex` VCF files. MH and ZI defined the graph constraints and edited the manuscript. MH and ZI designed, and BL modified, the genotyping model. All authors reviewed and approved the manuscript.

### Funding

BL is funded by an EMBL predoctoral fellowship. MH is funded by the Wellcome Trust/Newton Fund-MRC Collaborative Award [200205] and the Bill & Melinda Gates Foundation Trust [OPP1133541]. Open Access funding enabled and organized by Projekt DEAL.

### Availability of data and materials

`gramtools` is open-source under an MIT license and publicly available on Github [52] (https://github.com/iqbal-lab-org/gramtools). The version of `gramtools` used in this paper is archived on Zenodo [53].

We provide an open repository for reproducing all results in this study, available at https://github.com/iqbal-lab-org/paper_gramtools_nesting. The repository README provides all the instructions and commands to obtain the data and to re-run each part of the analysis (using Snakemake [54]). The data used are all openly available. Accessions for data deposited at the ENA are stored as tables, listed in https://github.com/iqbal-lab-org/paper_gramtools_nesting#input-data. All other data, as well as a software container, are available on zenodo at https://doi.org/10.5281/zenodo.5075458. All versions/commits of the software used in this study are frozen in the software container and can be found at https://github.com/iqbal-lab-org/paper_gramtools_nesting/blob/master/container/singu_def.def.

## Declarations

### Ethics approval and consent to participate

Not applicable.

### Competing interests

The authors declare that they have no competing interests.

### Author details

[1]EMBL-EBI, Hinxton, UK. [2]Nuffield Department of Medicine, University of Oxford, Oxford, UK.

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

## 