## [**Additional file 2** Review history · Genome Biology]

Review History

First round of review

Reviewer 1

Were you able to assess all statistics in the manuscript, including the appropriateness of statistical tests used? No, I do not feel adequately qualified to assess the statistics.

Were you able to directly test the methods? No

Comments to author:

This manuscript describes a new format for storing hierarchical data on both InDels and SNPs and a program, gramtools, that can generate files in that format. gramtools also produces standard VCF files that can be used by other, existing programs. The authors claim that hierarchical data containing both InDels and SNPs is not yet adequately handled by other software, and that this manuscript is a major innovation. gramtools is compared with other genome graph tools for accuracy, speed and computational requirements, and yielded higher accuracy because no other program handles both indels and SNPs, and with reasonable computational requirements.

Disclaimer: I am not a professional bioinformatician, nor have I tested the installation and use of GramTools. I cannot comment on ease of use of this program.

The manuscript uses two primary examples of possible uses of this approach, one with families of genes expressing antigens in *P. falciparum* and a second on the combination of regions of difference due to deletions and SNPs within those regions within *M. tuberculosis*. These are both well chosen because they illustrate the necessity for a hierarchical system that can encompass independent both large sequence variants and SNPs in a hierarchical system which allows more accurate alignments at the local level. What is lacking here is an estimate of the limits of this approach. Both sets of analyses are performed on a few thousand samples of genomes. Will gramtools be able to deal with 100,000s of genomes, which is already the norm in microbial genomics? And how does its speed relate to datasize? This needs to be addressed prior to publication of this work.

This manuscript reads a bit like a progress report. The authors have apparently been developing this methodology over time (citation 14), and have now reached a stage where they have both a format and a tool which can be formally described. However, the text is not explicit in this point, and it needs to be clarified.

The manuscript is written by computer scientists for computer scientists. If the approach is to gain broad use, then the manuscript would benefit from some minor changes,

1. A sketch or description of the format used used by VCF files and jVCF files.
2. improvements in some of the figures.

Fig. 1. Make the arrowheads in the subfigures such as Build and Genotype larger. I missed them totally at first. gramtools infers a reference genome for a sample. Does it provide an estimate of adequacy for that reference genome or complain if it cannot generate an adequate reference genome The

Fig. 2. Introduce discontinuities in the broken red lines. At first glance it looks as if these are alternative paths whereas they are intended to represent forbidden paths. Dotted red lines do not translate to forbidden paths unless they are broken. The figure legend refers to opening node and closing node but the particular nodes that fulfil these purposes are not specified. Include symbols showing opening and closing nodes in the figure.

Fig. 3. I could not understand the logic in parts c and d and the figure legend needs rewriting.

Fig. 4. The details are invisible on a paper printout of the manuscript. I suggest either simplifying the figure with larger elements or banishing it to supplementary or both.

Fig. 5. I could not understand this figure and it did not seem to relate to the contents in the text.

Fig. 6. this makes no sense at all.

Fig. 8a. Basically unintelligible.

Fig. 8b. The colours are particularly unfortunate in a printout, especially almost invisible light red on dark red to mark rectangles and the use of continual colour densities overlapping from values of 0 to -1 for the haplogroup.

The text would benefit from moving technical aspects to a special section in the manuscript and possibly moving some of the more arcane methods into Supplementary.

One of the goals of Genome Biology is to publish papers of broad interest to others in the field, or of outstanding interest to a broad audience of biologists. This paper has that potential but is written for a much smaller group of computer scientists. It would benefit from an attempt to make it comprehensible to the lay public.

Reviewer 2

Were you able to assess all statistics in the manuscript, including the appropriateness of statistical tests used? No, I do not feel adequately qualified to assess the statistics.

Were you able to directly test the methods? No

Comments to author:

Brice Letcher and colleagues present a new computational method (gramtools) for analyzing nested, hierarchical variation using short illumina reads. The method extends current state-of-the-art graph-based genome analysis and tackles the important problems of analyzing SNPs and structural variation together as well as representing variation between genomes (or across a population of genomes) in a way that is more interpretable for downstream analysis compared to current approaches.

Overall, I found the manuscript to be very well written. The concepts and approach are explained very clearly. I concur with the authors that better methods are needed to integrate SNP and structural variant genotyping and I like their emphasis on creating variant representations that are more amenable to downstream interpretation than the standard approaches currently used by graph genomes. This work describes some nice advances on the state of the art.

Major Comments

1. The manuscript starts out very strong, but I found the section covering Figs 5 and 6 less inspiring before the paper picked up steam again with Figs 7 and especially Fig 8. I might suggest shortening the middle part of the manuscript.

One thought would be to combine all four benchmarked genes into one figure (to replace Fig 5), perhaps leaving the results for each individual gene in the supplement if warranted.

Also noted in my minor comments below, it was a little unclear on first reading whether the PR is exactly the same as the gramtools genotypes for each sample (I decided it was). In this context, it seems like the main point of Fig. 5 should be that gramtools out-performs samtools and Cortex in this benchmark. Comparisons of samtools (or Cortex) running on the output (PR) of gramtools seems like a secondary point and less important. Worth including, but perhaps in a way that distracts less from the main result.

2. I find the section "Comparing gramtools personalised reference with best of reference panel" less compelling than other parts of the manuscript. I would recommend dropping it or shortening it significantly. Either that or you should better motivate why the reader should care about comparing to the best input from the graph.

In this context, I find Fig 6. not very informative. These differences in the scaled edit distance numbers are difficult to assess in isolation. Perhaps it would be better to show this as a scatter plot with an $x=y$ line drawn for reference. Is it the case that the differences are relatively small compared to the distance from either one to the truth?

3. Figure 8 is very nice.

It leaves me with a number of questions, though, and I think it would strengthen the paper if you were to devote some more space to analyzing these results. If the authors feel this is beyond the scope of the paper, I won't quibble, but any added discussion on some of the following questions would be welcome:

a) Is there some basis for determining that the two divergent forms are delineated by the red arrow on the left?

Are there some specific markers that differentiate the forms? If so, perhaps you could indicate these distinguishing markers on the plot.

b) Is there some specific reason for highlighting the one divergent segment on the top (as opposed to the other green segments)?

c) You highlight two "clear cases" of apparent recombination, but it looks by eye like there are several more. Is that incorrect?

If these other cases do not suggest recombination, it would be interesting to explain why.

d) Have you reconstructed a phylogenetic tree from this analysis? It would be quite an interesting addition.

I feel like this example illustrates the power of the methods developed by the authors and including more biology and analysis would serve to highlight the impact of the work.

4. *P. Falciparum* and *M. Tuberculosis* are haploid. Has this method been applied to diploid genomes?

It would be nice to include a discussion of whether gramtools can be immediately applied to diploid genomes in its present form or whether this would require significant enhancements.

Minor Comments

1. In a couple of places the authors mention that there are no existing solutions for analyzing SNPs with SVs (e.g. page 2, line 44 and page 3, line 9).

Although it is older and not directly applicable (because it is not sequencing based), the Birdsuite software (<https://pubmed.ncbi.nlm.nih.gov/18776909/>) did analyze SNPs within the context of SVs using genotyping arrays. Perhaps referencing this would add historical interest.

2. Page 5, line 47. Wording doesn't scan well. "which ones variants fall on"

3. Page 6, bottom: It is a little confusing how you can get more sites from the nested graph.

Is this an issue of representation and how one counts "sites"? E.g. like splitting an MNP or a complex low-complexity site (give example) into many smaller alleles? If this is related to Fig 4 (more nodes in (a) relative to (b)) then referring to this figure this would be an excellent way to explain.

4. Fig 5 and supp figs 5-7. I am a little unsure whether the personal reference (PR) for a sample is exactly the same as the gramtools calls for the sample or not. I assume that they are.

5. The legend for Fig 5 should maybe be reworded for clarity: "the x-axis is the scaled edit distance (edit distance divide by the length of the DBLMSP2 gene) to the true sequence determined from a

high-quality PacBio assembly ..." Also "The two histograms in the left-hand column ... are to the truth data for each sample".

6. Supp Fig. 6. I think this should say "all but one" (not "all except two").

7. The legends for Supp Figs 5-7 should be more detailed and self-contained. I would suggest repeating the information from the main Fig 5 about what is being compared. in Supp Fig 5 and then Supp Figs 6 and 7 could say "Similar analysis as in Supp Fig 5").

8. Fig 6. If this figure is kept, I would like the legend to also be more self-contained, describing what is being compared (the gene is DBMLSP2, the truth is the pacbio assembly, etc.). This information is in the text, but should be in the legend to make the figure self-contained in case the reader is going straight to the figures.

9. Page 11, performance table.

It would be easier to read the genotype/RAM column if you didn't put the second value in parens (because Mb in the title is in parens) but instead used square brackets or something, e.g. "605 [158]".

10. Page 11, performance table legend.

In the legend text, "else temporary disk use exceeded...". Reword for clarity.

If I got this right, do you mean something like "Index: building genome graph from input VCF. (vg graph was pruned to reduce graph complexity ...)"

11. Page 11, line 45. Perhaps this small paragraph should be tacked on as a single sentence on line 32 (after the reference to minimap2). Something like "Similar results were obtained using bowtie2 instead of minimap2 - see supp Figure 13, methods".

12. Page 11, line 37: I don't like lumping the reference genome into this comparison. It is a fine baseline, but "assume all sites match the reference" is not a valid genotyping method. I would suggest rewording to make it clear that the reference is just used as a baseline. Similarly, I might suggest dropping "all tools perform better than using the reference sequence alone". One would hope so!

13. Page 11, bottom: "We are for example aware of gramtools' much lower mapping speed to the human genome ..." This sounds apologetic, but it isn't clear to me that it needs to be. For example, you could claim that in this bacterial benchmark you are 4-8x faster than vg/GraphTyper2, but then discuss that performance may differ in different benchmarks and that gramtools is slower when mapping to the human genome.

14. Fig 7. I think you should expand the legend to be more specific and more precise.

I'm not sure what "sequences" refers to here. If I understood the analysis correctly, perhaps something like "The curves are the cumulative frequencies of edit distance between inferred sequences ... at 73 large deletion sites in 17 samples compared to truth assemblies from PacBio long reads".

15. Fig 8. In panel (b), the x axis is not labeled (or explained).

16. Methods. Page 17, line 21. I'm not sure this paragraph and some of the following is exactly right.

Specifically, on line 22, you say that "S is non-empty and totally ordered", but you don't specify what this total order is. If you intend to use the partial order defined by the edges of the DAG itself, then I don't believe $c(v)$ has to be unique (unless the DAG is a lattice, which you have not specified).

After thinking about this a bit, I think what you are intending here is that gramtools supports DAGs such that there exists a total order T that is a topological sort of the DAG and that the DAG must have certain properties with respect to T , in particular that T is a balanced-bracket ordering of the DAG. If you reframe it this way, then it is also important to distinguish between whether T exists and whether you have an algorithm to find a suitable total order T if it exists.

As an aside, although I wasn't able to work this out in detail, I wonder whether the set of NDAGs supported by gramtools are exactly those DAGs that are (finite, and therefore complete) lattices. I wasn't able to sketch a proof (or counter-example) in the margins, but the idea would be that $c(v)$ would be the meet of the successors of v and $o(v)$ would be the join of the predecessors of v . You would then have to prove that every lattice L has a total order T such that T is a balanced-bracket ordering for L , ideally by showing a way to construct T . And conversely show that every DAG D , if there exists a total order T that is a balanced-bracket ordering for D , then D is a lattice.

17. Page 18, line 7: I think you mean all walks from s_1 to s_2 (not "all walks from s_1 ").

18. Page 23, line 40: Since there are only 5 manual validations, it wouldn't be bad to include all 5 plots in the supplement as you did for the example in Supp Fig 11.

19. Supp Fig 12 does not seem to be referenced from the text.

20. Supplement

It is a little weird to have the jVCF specification stuck into the middle of the supplement.

The reader has to scroll through it to find the figures referenced by the manuscript. Does it really need to be in the supplement if it is available online? If it is attached as a supplement, I would suggest making it a different supplementary file or putting it at the end of the supplement in a separate section.

Reviewer 3

Were you able to assess all statistics in the manuscript, including the appropriateness of statistical tests used? Yes, and I have assessed the statistics in my report.

Were you able to directly test the methods? Yes

Comments to author:

In this paper by Letcher et al. a novel approach called gramtools is introduced to model genomes as directed acyclic graph. They proposed approach allowed successive hierarchical subgraphs which gives the ability to genotype and represent regions of genomes with high variability between samples.

Major Comments:

1. The results in Figure 7 is quite good. However, I think the paper would benefit from more discussion and breakdown of their genotyping ability especially based on different types of variants (and or size of events)
2. Although the authors argue their method is not designed for studying human genome (long genome and low variability), however, I still think some sort of results with human genomes would be interesting (even if only for one chromosome such as chr22).
3. I think the authors can discuss if the proposed approach can be incorporate with imputation techniques.

Response to Reviewers

We would like to thank the reviewers for their consideration of our manuscript. In addressing the issues they raised, we feel we have produced a much clearer and more focussed study. We give below a detailed response to review, but at a high level, the key observations of the reviewers were that we needed to improve the accessibility of the exposition to a wider biological audience, and to clarify the narrative flow so that the reader was more naturally led to the novelty of the results, rather than being bogged down in details up-front. We took these to heart, and made the following changes.

- We have gone through the whole manuscript with a view to improving accessibility of the text. The idea of a pan-genome is widely known, although it means slightly different things in different communities (microbiology, plant science, human genetics), but a major sticking point is how this rather abstract idea can be used for the analysis of genetic variation. This manuscript does not claim to offer a final solution, but rather seeks to resolve the technical problem of how to co-analyse variants on different genetic backgrounds, and those “underneath” large deletions. We therefore have sought to keep the line of argument clear, and focussed on the key aspects, and highlighted the key biological benefits.
- Some sections in the original submission which really belonged in the Methods or Supplementary were moved there. The figure describing nested genotyping was moved to the Methods. Figure 5 was simplified and details moved to the Supplementary Material. Figure 6 which was making a rather detailed point about approximating a genome as a recombinant, was also moved to the Supplementary Material. There has been some figure renumbering as a result, which we summarise below in a table.
- The key results sections, showing how our method allows genotyping at loci where long deletions overlap SNPs (in *M. tuberculosis*) and allows access to SNPs on top of multiple alternate haplotypes (in *P. falciparum*) have been clarified. The major figure of the paper shows how, in a cohort of 706 *P. falciparum* genomes from Ghana, Laos and Cambodia, around half of the genomes have a rich pattern of SNPs which is invisible to single-reference based methods. Both reviewers expressed reservations with the original version of the figure, and we have substantially improved it, and the associated description.

We give below a table showing how old figure numbers correspond to new, and then a detailed response to the reviews. In the response, we refer to page and line numbers of a revised pdf document containing the differences as compared to the previously reviewed version.

Figures

Table explaining the correspondence between old/new figures below.

Old figure number	New figure number
-------------------	-------------------

Fig. 3	Fig. 9 (moved to the methods section)
-	Fig. 3 (new figure)
Fig. 6	Supp. fig. 11
Fig. 7	Fig. 6
Fig. 8	Fig. 7
Fig. 9	Fig. 8
-	Supp. fig.4, 5, 7-16, 19-20 (new figures)

Unchanged figures are not mentioned (eg Fig. 1 in reviewed manuscript is still Fig. 1 in revised manuscript).

Reviewer #1:

This manuscript describes a new format for storing hierarchical data on both InDels and SNPs and a program, gramtools, that can generate files in that format. gramtools also produces standard VCF files that can be used by other, existing programs. The authors claim that hierarchical data containing both InDels and SNPs is not yet adequately handled by other software, and that this manuscript is a major innovation. gramtools is compared with other genome graph tools for accuracy, speed and computational requirements, and yielded higher accuracy because no other program handles both indels and SNPs, and with reasonable computational requirements.

Disclaimer: I am not a professional bioinformatician, nor have I tested the installation and use of GramTools. I cannot comment on ease of use of this program.

The manuscript uses two primary examples of possible uses of this approach, one with families of genes expressing antigens in *P. falciparum* and a second on the combination of regions of difference due to deletions and SNPs within those regions within *M. tuberculosis*. These are both well chosen because they illustrate the necessity for a hierarchical system that can encompass independent both large sequence variants and SNPs in a hierarchical system which allows more accurate alignments at the local level. What is lacking here is an estimate of the limits of this approach. Both sets of analyses are performed on a few thousand samples of genomes. Will gramtools be able to deal with 100,000s of genomes, which is already the norm in microbial genomics? And how does its speed relate to dataset size? This needs to be addressed prior to publication of this work.

We thank the reviewer for highlighting this point, which was indeed a gap. The question can be addressed at two levels. First, can gramtools be used practically, for reasonable sizes of dataset? Second, how does it scale theoretically - this aspect can be complex as computer science methods for describing complexity can sometimes focus on theoretical worst-case performance, and so needs to be backed up by empirical data. In the discussion section, we have now added the following text addressing these.

<Page 17, line 31>: “Choice of graph model also affects the computational performance of read mapping. Data structures supporting linear-time exact matching exist for a restricted class of graphs (see Wheeler graphs [33], which include De Bruijn graphs). In gramtools, graphs do not have this property, and we use a data structure (the vBWT) with run time and memory use that can scale exponentially with genome size and density of stored variation. In practice however, gramtools shows good computational performance on the *P. falciparum* and *M. tuberculosis* graphs used in this study. Here we analyse variation from a few thousand samples and a small number of regions, but gramtools has also successfully been used to genotype 70,000 (whole genome) *M. tuberculosis* samples in a graph containing 1.25 million variants (one variant every 3.5 bases)[50]. We are also, however, aware of gramtools’ very low mapping speed on the human genome [31]. More work is required in the pangenomics field to understand the different real-world performance challenges of repetitiveness (*P. falciparum* is much more repetitive than human), genome size (microbes are tiny but diverse) and amount, type and density of variation.”

We do not go into more detail in that paragraph on performance in that large *M. tuberculosis* cohort, since it would divert the narrative, and is the subject of another study, but we think this addresses the issue raised by the reviewer.

This manuscript reads a bit like a progress report. The authors have apparently been developing this methodology over time (citation 14), and have now reached a stage where they have both a format and a tool which can be formally described. However, the text is not explicit in this point, and it needs to be clarified.

We apologise for the lack of clarity. It is common practice for one paper to introduce an algorithm and describe its characteristics, and for a later paper to then build on this to produce a workable tool for general use. This is what has happened here. We have modified the text as follows

<Page 4, line 44>

“The vBWT algorithm was introduced with a proof-of-concept implementation in [12]”

<Page 6, line 9>

“The original vBWT implementation was slow and did not support nesting at all [12]. In this paper we introduce our nesting implementation, and have optimised the codebase to improve mapping, coverage recording, and genotyping.”

The manuscript is written by computer scientists for computer scientists. If the approach is to gain broad use, then the manuscript would benefit from some minor changes,

We thank the reviewer for highlighting this issue, which has led us to review the whole paper and rewrite some sections for clarity.

1. A sketch or description of the format used used by VCF files and jVCF files.

Thank you for this suggestion, we have added a figure illustrating VCF and jVCF on an example graph, in the “jVCF output format” section. This is the new Figure 3 on Page 8.

2. improvements in some of the figures.

Fig. 1. Make the arrowheads in the subfigures such as Build and Genotype larger.

Done

I missed them totally at first. gramtools infers a reference genome for a sample. Does it provide an estimate of adequacy for that reference genome or complain if it cannot generate an adequate reference genome The

We have added the following sentence <Page 6, line 21>:

“The personalised reference (PR) genome inferred by gramtools is the path obtained by taking the maximum-likelihood call at each variant site, and the genotype confidence at each site provides a measure of the adequacy of the inferred PR. For example, a stretch of low-confidence calls suggests no close path in the graph was found, or that no reads mapped in this region”

Our simulations and empirical measurements also directly measure the adequacy of the approach.

Fig. 2. Introduce discontinuities in the broken red lines. At first glance it looks as if these are alternative paths whereas they are intended to represent forbidden paths. Dotted red lines do not translate to forbidden paths unless they are broken. The figure legend refers to opening node and closing node but the particular nodes that fulfil these purposes are not specified. Include symbols showing opening and closing nodes in the figure.

We have added discontinuities for the forbidden paths and added symbols for the opening and closing nodes. We believe the figure is now clear.

Fig. 3. I could not understand the logic in parts c and d and the figure legend needs rewriting.

We no longer feel this figure belongs in the main text, and have moved it to the Methods, now Figure 9. We have rewritten the figure legend to make parts c and d more understandable.

Fig. 4. The details are invisible on a paper printout of the manuscript. I suggest either simplifying the figure with larger elements or banishing it to supplementary or both.

We have increased the font size and enlarged the arrows to improve readability.

Fig. 5. I could not understand this figure and it did not seem to relate to the contents in the text.

The idea of figure 5 is to benchmark how close to the truth one can get using gramtools and other single-reference tools. We measure this distance from the truth very simply, by aligning the truth sequence and the inferred sequence and counting mismatches and indels, and then rescaling by gene length to allow comparison between genes. We then simply plot a

histogram showing, on our test set, how close the results are to zero, which is the ideal result (zero differences between truth and inference). We show in one panel, as a baseline, how close the reference genome is to all the samples - this histogram is the most spread out, and essentially measures diversity in our data. We then compare the histograms for samtools ("the canonical variant caller", which is essentially the baseline for the last 10 years), cortex (the only variant caller which has previously been able to access these genes, in Miles et al 2016) and gramtools. As desired, the gramtools histogram is more compressed to the left than all the others. We have simplified Fig. 5 to show only this main result, which is that gramtools outperforms single-reference callers, and moved the more technical exposition to the supplementary. The section of text it refers to is on Page 10, line 20.

Fig. 6. this makes no sense at all.

First of all we agree and apologise, the figure was not sufficiently clear. We have completely changed how the data is now presented, which we will describe momentarily. However, in addition we no longer feel this figure belongs in the main text, and have moved it to the supplementary material, now denoted Supplementary Figure 11. The new version of this figure has 4 panels, one per gene, each of which is a scatter plot, with each dot representing the gene sequence for one of the 14 *P. falciparum* samples. One axis shows the distance of the true gene sequence from the gramtools inferred-reference, and the other shows the distance of the true gene sequence from the nearest input allele when building the graph (ie the closest allele in the reference panel). By looking at whether points are above the line $y=x$, one is able to infer whether or not gramtools found a new sequence (not in the input data) which is close to the sample. We believe this is a much more intuitive presentation, the idea for which we owe to Reviewer #2 (see below), who also suggested we move it to the supplementary (as we have done).

Fig. 8a. Basically unintelligible.

We have dramatically modified this figure to improve clarity, and as a result Fig 8a (which was a cartoon designed to aid the understanding of Figure 8b) no longer exists, there is just a single panel, which (due to figure renumbering) is now Figure 7.

Fig. 8b. The colours are particularly unfortunate in a printout, especially almost invisible light red on dark red to mark rectangles and the use of continual colour densities overlapping from values of 0 to -1 for the haplogroup.

We agree, and have modified the palette accordingly. The new Figure 7 (corresponds to previous Fig 8b) is also clearer and we feel more intuitive, no longer relying on the reader concentrating through an initial didactic panel (previous Fig 8a). The goal of this figure is to show clearly how gramtools reveals patterns of genetic variation on different, highly diverged, genetic backgrounds, and in fact show that for around half of the 706 genomes, this variation would be invisible if working from the standard reference genome.

Traditionally, with a linear reference and a set of (say) 1000 biallelic SNPs, each sample's genotypes can be coded as 1000 zeroes or ones. Thus when looking at a cohort of genomes, their genotypes can be shown as a heatmap, drawing black for zero and white for one, for example. Haplotype patterns then become easily visible to the eye. We want to do the same, despite the fact that we have a graph instead of a linear genome. Since the graph

is directed, with a clear left-right orientation, we can linearise it while preserving that left-right sense. We need more colours than black and white, but nevertheless we produce a heatmap that does indeed show haplotype patterns, and also encodes information about the graph.

Our figure therefore has the heatmap at the centre, bracketed on either side by annotation: on the left a hierarchical clustering tree shows that the sequences split into two deeply diverged types, one of which itself has two subforms, and a vertical strip shows which samples come from Cambodia, Laos or Ghana. At the bottom right, we have added a blow-out which recapitulates the heatmap but divides it into coloured blocks showing key haplotype patterns, which the reader can then compare with the detailed patterns on the heatmap. This allows the user to more easily see the haplotypes. We also add, on the right, annotation strips showing the two diverged forms of the gene, and how they can be shown as combinations of haplotypes. In this way, it becomes extremely easy to see two sets of genomes which are clearly inter-form recombinants, and that they both can be found in all 3 countries. We believe this is a much clearer representation. The information in the previous Figure 8a, about how the graph gets linearised, is shown in a blow-up above the heatmap.

The text would benefit from moving technical aspects to a special section in the manuscript and possibly moving some of the more arcane methods into Supplementary.

One of the goals of Genome Biology is to publish papers of broad interest to others in the field, or of outstanding interest to a broad audience of biologists. This paper has that potential but is written for a much smaller group of computer scientists. It would benefit from an attempt to make it comprehensible to the lay public.

We completely agree, and have reworked the paper to achieve this end. We have significantly reduced the amount of technical exposition in the Results section of the paper. Fig.3, describing the algorithm for nested genotyping, was moved to the Methods, and Fig.6, which was technical and not as relevant as the other Figures, was moved to the Supplementary Material. We have also simplified Figure 5 and the accompanying text, and we have added a new figure giving an overview of VCF and jVCF. Finally, the reworking of Figure 8 (now Figure 7) allows the key messages to be absorbed without studying the caption in detail. We believe this makes the paper much more comprehensible to a broader audience of biologists and bioinformaticians.

Reviewer #2:

Brice Letcher and colleagues present a new computational method (gramtools) for analyzing nested, hierarchical variation using short illumina reads. The method extends current state-of-the-art graph-based genome analysis and tackles the important problems of analyzing SNPs and structural variation together as well as representing variation between genomes (or across a population of genomes) in a way that is more interpretable for downstream analysis compared to current approaches.

Overall, I found the manuscript to be very well written. The concepts and approach are explained very clearly. I concur with the authors that better methods are needed to integrate SNP and structural variant genotyping and I like their emphasis on creating variant representations that are more amenable to downstream interpretation than the standard

approaches currently used by graph genomes. This work describes some nice advances on the state of the art.

We thank the reviewer for these comments, and their extremely detailed review.

Major Comments

1. The manuscript starts out very strong, but I found the section covering Figs 5 and 6 less inspiring before the paper picked up steam again with Figs 7 and especially Fig 8. I might suggest shortening the middle part of the manuscript.

We agree with your points below, so have shortened and simplified the section covering Fig.5, and moved the section covering Fig. 6 to the Supplementary.

One thought would be to combine all four benchmarked genes into one figure (to replace Fig 5), perhaps leaving the results for each individual gene in the supplement if warranted.

We agree, and have combined all four genes in one figure, replacing Fig. 5. As recommended, the per-gene results are now in the supplementary material.

Also noted in my minor comments below, it was a little unclear on first reading whether the PR is exactly the same as the gramtools genotypes for each sample (I decided it was).

We apologise for the lack of clarity. You were indeed correct. We now have the following descriptive text on Page 5 for Figure 1:

“Genotyping consists of calling alleles at each variant site and thereby inferring a haploid personalised reference genome for a sample.”

As well as this on Page 6, line 21:

““The personalised reference (PR) genome inferred by gramtools is the path obtained by taking the maximum-likelihood call at each variant site, and the genotype confidence at each site provides a measure of the adequacy of the inferred PR.”

In this context, it seems like the main point of Fig. 5 should be that gramtools out-performs samtools and Cortex in this benchmark. Comparisons of samtools (or Cortex) running on the output (PR) of gramtools seems like a secondary point and less important. Worth including, but perhaps in a way that distracts less from the main result.

The main point is indeed that gramtools outperforms the single-reference variant callers. To focus on this point, we removed samtools/cortex against the personalised reference (PR), and clarified the text (starting Page 9, line 11). (The results for each gene and samtools/cortex against the PR were moved to the Supplementary (text on Page 9, figure numbers 7-10))

2. I find the section "Comparing gramtools personalised reference with best of reference panel" less compelling than other parts of the manuscript. I would recommend dropping it or

shortening it significantly. Either that or you should better motivate why the reader should care about comparing to the best input from the graph.

We agree that this section does not belong to the main results, and have moved it to Supplementary Figure 11. In terms of motivation, we have the following sentence in the Supplementary text now:

“One advantage of using graphs for genotyping is that the inferred sample’s path through the graph can be a recombinant of the sequences used to build the graph.”

In this context, I find Fig 6. not very informative. These differences in the scaled edit distance numbers are difficult to assess in isolation. Perhaps it would be better to show this as a scatter plot with an $x=y$ line drawn for reference. Is it the case that the differences are relatively small compared to the distance from either one to the truth?

Thank you for this suggestion. We have redrawn this plot as a scatterplot (Supp. Fig. 11), so that the reader can also see the distance between gramtools/the closest input sequence and the truth, and it is a considerable improvement.

3. Figure 8 is very nice.

It leaves me with a number of questions, though, and I think it would strengthen the paper if you were to devote some more space to analyzing these results. If the authors feel this is beyond the scope of the paper, I won’t quibble, but any added discussion on some of the following questions would be welcome:

Thank you. We have absorbed these suggestions, along with the strong message from Reviewer 1 that this figure was too difficult to understand, and have reworked the figure considerably. (We mention also that this Figure is now numbered Figure 7).

We first repeat here our description of the new figure which we gave above for Reviewer 1:

“The new Figure 7 (corresponds to previous Fig 8b) is also clearer and we feel more intuitive, no longer relying on the reader concentrating through an initial didactic panel (previous Fig 8a). The goal of this figure is to show clearly how use of a gramtools reveals patterns of genetic variation on different, highly diverged, genetic backgrounds, and in fact show that for around half of the 705 genomes, this variation would be invisible if working from the standard reference genome.

Traditionally, with a linear reference and a set of (say) 1000 biallelic SNPs, each sample’s genotypes can be coded as 1000 zeroes or ones. Thus when looking at a cohort of genomes, their genotypes can be shown as a heatmap, drawing black for zero and white for one, for example. Haplotype patterns then become easily visible to the eye. We want to do the same, despite the fact that we have a graph instead of a linear genome. Since the graph is directed, with a clear left-right orientation, we can linearise it while preserving that left-right sense. We need more colours than black and white, but nevertheless we produce a heatmap that does indeed show haplotype patterns, and also encodes information about the graph.

Our figure therefore has the heatmap at the centre, bracketed on either side by annotation: on the left a hierarchical clustering tree shows that the sequences split into two deeply diverged types, one of which itself has two subforms, and a vertical strip shows which samples come from Cambodia, Laos or Ghana. At the bottom right, we have added a

blow-out which recapitulates the heatmap but divides it into coloured blocks showing key haplotype patterns, which the reader can then compare with the detailed patterns on the heatmap. This allows the user to more easily see the haplotypes. We also add, on the right, annotation strips showing the two diverged forms of the gene, and how they can be shown as combinations of haplotypes. In this way, it becomes easy to see two sets of genomes which are clearly inter-form recombinants, and that they both can be found in all 3 countries. We believe this is a much clearer representation. The information in the previous Figure 8a, about how the graph gets linearised, is shown in a blow-up above the heatmap.”

In response to your suggestions that it would be good to expand a bit more on the biology, we have done this in two ways. Firstly, we have worked hard to make it much more intuitive how to see the haplotypes and recombinants in the heatmap. Second, we have expanded the text in the body of the paper. The new text reads as follows:

<Page 16, line 1> “Fig. 7 shows a matrix of calls in each sample at each variant site inside the DBL domain of DBLMSP2, known to be dimorphic [32]. The tree on the left depicts a hierarchical clustering of the sample haplogroups. Its most basal split distinguishes two forms of the domain (form 1 (dark pink) and form 2 (light pink) on the right of the heatmap), with an average scaled edit distance between the two forms of 16.8% compared to within-form distances of 1.4% and 4.8%. gramtools can thus recover two divergent forms, as expected given the known dimorphism. Building a phylogenetic tree of the sequences confirmed the presence of the two forms and showed high concordance with the clustering tree in how samples are assigned to each form (Supp. Fig. 20).

gramtools also produces calls in nested sites, shown as dark bars at the top row of the matrix. As an example, we provide an illustrative blowout at the top of the heatmap of a region where variants are nested and occur on different sequence backgrounds. The graphs illustrate how the nested sites are called mutually exclusively: when one of the gene forms has a genotype call, the other form receives a null call (shown as black-coloured cells), and vice-versa. This is essentially showing that there are SNPs on both genetic backgrounds (the dimorphic types) and gramtools is genotyping variants irrespective of background.

Interestingly, the heatmap in Fig.7 also shows, for the first time to our knowledge, clear evidence of recombination between the two forms of DBLMSP2. At the bottom-right of the figure, we show a blowout of the matrix that is coloured by broad haplotype patterns, with form 1 being predominantly yellow followed by green and form 2 being predominantly blue or grey followed by red. This reveals two sets of samples (labelled A,B by the blowout) which are inter-form recombinants. Those labelled A have a yellow haplotype (form 1) followed by red (form 2), and those labelled B have blue (form 2) followed by green (form 1). The leftmost column of the matrix is coloured by the country of origin of each sample. Strikingly, for each group of samples (ie A and B) almost identical recombinants exist across each of Ghana, Laos and Cambodia. We do not yet know if these inter-form recombinants derive from a single recombination event (either ancestral to the samples or transmitted across countries), or if this recombination event has occurred multiple times independently.

Finally, we indicate, using a red asterisk on the right of the tree, the closest clade to the reference genome sequence 3D7. Leveraging gramtools’ graph model and jVCF output, we

can move beyond a 3D7-only reference-based analysis that would neither genotype nested variants on the two different backgrounds nor reveal the inter-form recombinants.”

a) Is there some basis for determining that the two divergent forms are delineated by the red arrow on the left?

We have now added a hierarchical clustering tree to the figure, showing how the alleles come from two deeply diverged types. We quantify this in the text:

<Page 16, line 4> “Its most basal split distinguishes two forms of the domain (form 1 (dark pink) and form 2 (light pink) on the right of the heatmap), with an average scaled edit distance between the two forms of 16.8% compared to within-form distances of 1.4% and 4.8%. “

Are there some specific markers that differentiate the forms? If so, perhaps you could indicate these distinguishing markers on the plot.

There are form-informative markers, but these are spread across the lengths of the haplotypes (this can be seen from the heatmap haplotype patterns), and it would not be very helpful to show them. We did make an effort to see what the heatmap would look like if displaying this information in a track at the top, but there were so many, we found it simply added too much information to the figure.

b) Is there some specific reason for highlighting the one divergent segment on the top (as opposed to the other green segments)?

We highlight this segment as it most clearly shows variation being called on different backgrounds. To make this clear, we have updated the figure to illustrate the situation using a cartoon blowout, and modified the caption as follows:

<Page 32> “As a clarifying example, the blowout at the top shows how two incompatible nested SNPs on different backgrounds are displayed as either dark-then-light-blue, or light-then-dark blue blocks in the heatmap, with the incompatible haplotype shaded black (null call).”

c) You highlight two "clear cases" of apparent recombination, but it looks by eye like there are several more. Is that incorrect?

If these other cases do not suggest recombination, it would be interesting to explain why.

This is not incorrect, but other apparent recombinations are either not between the two forms, or less clear visually. We have updated the figure to really focus on the two ‘clear cases’ of inter-form recombinations so that the reader can easily get what recombination patterns look like.

d) Have you reconstructed a phylogenetic tree from this analysis? It would be quite an interesting addition.

We have. The phylogenetic tree distinguishes the two forms almost identically to the clustering tree (which we have added to Figure 7); we state this in the text and added the tree as Supplementary Figure 20.

I feel like this example illustrates the power of the methods developed by the authors and including more biology and analysis would serve to highlight the impact of the work.

Thank you, we agree. We hope our new figure and observations have highlighted clearly that in this biologically important surface antigen, using a single reference hides essentially half of the haplotypes (the top half of the heatmap) and some very interesting inter-form recombination. We also show, very surprisingly, that these recombinants can be found in 3 different countries from Africa and South East Asia. This is an unexpected (and striking) result. There are natural questions to follow up but they would significantly divert from the main message of the paper, and we plan to investigate in a further study.

4. *P. Falciparum* and *M. Tuberculosis* are haploid. Has this method been applied to diploid genomes?

It would be nice to include a discussion of whether gramtools can be immediately applied to diploid genomes in its present form or whether this would require significant enhancements.

We have not applied gramtools to diploid genomes, but gramtools can genotype diploid samples. The current limitation for diploid organisms is that two personalised reference (PR) genomes are produced, one for each chromosome, and heterozygous variants are unphased (thus alleles are randomly assigned to the two PRs). We now state this in the paper:

<Page 6, line 26> “While the genotyping model can handle both haploid and diploid cases, in the diploid case two unphased PRs are produced (as was done in [3]) whereby the two alleles at heterozygous sites get randomly allocated to each. In this paper we evaluate gramtools on haploid organisms only.”

Minor Comments

1. In a couple of places the authors mention that there are no existing solutions for analyzing SNPs with SVs (e.g. page 2, line 44 and page 3, line 9).

Although it is older and not directly applicable (because it is not sequencing based), the Birdsuite software (<https://pubmed.ncbi.nlm.nih.gov/18776909/>) did analyze SNPs within the context of SVs using genotyping arrays. Perhaps referencing this would add historical interest.

Thank you, we were unaware of this software and now reference it on Page 3, line 34.

2. Page 5, line 47. Wording doesn't scan well. "which ones variants fall on"

We replaced this with “placing variants based on what haplogroup they fall on”.

3. Page 6, bottom: It is a little confusing how you can get more sites from the nested graph. Is this an issue of representation and how one counts "sites"? E.g. like splitting an MNP or a complex low-complexity site (give example) into many smaller alleles? If this is related to Fig

4 (more nodes in (a) relative to (b)) then referring to this figure this would be an excellent way to explain.

This is indeed related to Fig.4, and we have updated the text to refer to Fig. 4. We also now explain that the nested graphs allow for SNPs to occur on the different sequence backgrounds that exist in these two genes (DBLMSP and DBLMSP2).

4. Fig 5 and supp figs 5-7. I am a little unsure whether the personal reference (PR) for a sample is exactly the same as the gramtools calls for the sample or not. I assume that they are.

This is indeed the case, and as mentioned above we have added a sentence in Fig.5's caption, and also in the section "Genotyping nested genome graphs" (Page 6, line 21), to make it clear.

5. The legend for Fig 5 should maybe be reworded for clarity: "the x-axis is the scaled edit distance (edit distance divide by the length of the DBLMSP2 gene) to the true sequence determined from a high-quality PacBio assembly ..." Also "The two histograms in the left-hand column ... are to the truth data for each sample".

We reworded the legend of Fig. 5 accordingly.

6. Supp Fig. 6. I think this should say "all but one" (not "all except two").

Thank you for spotting this discrepancy, it made us realise there are two imperfectly resolved sequences by gramtools, but the plot was wrongly showing only one due to an error. We corrected this.

7. The legends for Supp Figs 5-7 should be more detailed and self-contained. I would suggest repeating the information from the main Fig 5 about what is being compared. In Supp Fig 5 and then Supp Figs 6 and 7 could say "Similar analysis as in Supp Fig 5").

We have implemented this.

8. Fig 6. If this figure is kept, I would like the legend to also be more self-contained, describing what is being compared (the gene is DBMLSP2, the truth is the pacbio assembly, etc.). This information is in the text, but should be in the legend to make the figure self-contained in case the reader is going straight to the figures.

This figure has been moved to the supplementary, combined with the other three genes and we have made a self-contained legend.

9. Page 11, performance table.

It would be easier to read the genotype/RAM column if you didn't put the second value in parens (because Mb in the title is in parens) but instead used square brackets or something, e.g. "605 [158]".

Agreed and changed.

10. Page 11, performance table legend.

In the legend text, "else temporary disk use exceeded...". Reword for clarity.

If I got this right, do you mean something like "Index: building genome graph from input VCF. (vg graph was pruned to reduce graph complexity ...)"

We reworded this part accordingly.

11. Page 11, line 45. Perhaps this small paragraph should be tacked on as a single sentence on line 32 (after the reference to minimap2). Something like "Similar results were obtained using bowtie2 instead of minimap2 - see supp Figure 13, methods".

We implemented this suggestion.

12. Page 11, line 37: I don't like lumping the reference genome into this comparison. It is a fine baseline, but "assume all sites match the reference" is not a valid genotyping method. I would suggest rewording to make it clear that the reference is just used as a baseline. Similarly, I might suggest dropping "all tools perform better than using the reference sequence alone". One would hope so!

The reference sequence is indeed a baseline and not a genotyping method. We have made this clear (see paragraph Page 13, line 22), and removed "all tools perform better than using the reference sequence alone".

13. Page 11, bottom: "We are for example aware of gramtools' much lower mapping speed to the human genome ..." This sounds apologetic, but it isn't clear to me that it needs to be. For example, you could claim that in this bacterial benchmark you are 4-8x faster than vg/GraphTyper2, but then discuss that performance may differ in different benchmarks and that gramtools is slower when mapping to the human genome.

We appreciate this suggestion! We rewrote the text in a less apologetic tone, as follows:

<Page 14, line 10> "In terms of computational performance, gramtools processed the most reads per

CPU second while using comparable amounts of RAM on this dataset (Table 1).

A bottleneck in vg is temporary disk use, exceeding 500 Gigabytes without pruning the graph to remove densely clustered variation. For GraphTyper2, we include counting a separate mapping step to the reference genome (with bowtie2) as it is a prerequisite to genotyping (for vg and gramtools, performance includes mapping reads to genome graphs before genotyping). While gramtools' mapping and genotyping is 4 to 8 times faster than vg and GraphTyper2 in this benchmark, we are also aware of gramtools' much lower mapping speed to the human genome [31]. Computational performance depends on the genome and variants under analysis and on the genome graph approach; we consider these further in the Discussion."

And in the discussion we say

<Page 17, line 31> "Choice of graph model also affects the computational performance of read mapping. Data structures supporting linear-time exact matching exist for a restricted class of graphs (see Wheeler graphs [33], which include De Bruijn graphs). In gramtools, graphs do not have this property, and we use a data structure (the vBWT) with run time and memory use that can scale exponentially with genome size and density of stored variation. In practice however, gramtools shows good computational performance on the *P. falciparum* and *M. tuberculosis* graphs used in this study. Here we analyse variation from a few thousand samples and a small number of regions, but gramtools has also successfully been used to genotype 70,000 (whole genome) *M. tuberculosis* samples in a graph containing 1.25 million variants (one variant every 3.5 bases). We are also, however, aware of gramtools' very low mapping speed on the human genome [31]. More work is required in the

pangenomics field to understand the different real-world performance challenges of repetitiveness (*P. falciparum* is much more repetitive than human), genome size (microbes are tiny but diverse) and amount, type and density of variation.”

14. Fig 7. I think you should expand the legend to be more specific and more precise. I'm not sure what "sequences" refers to here. If I understood the analysis correctly, perhaps something like "The curves are the cumulative frequencies of edit distance between inferred sequences ... at 73 large deletion sites in 17 samples compared to truth assemblies from PacBio long reads".

Thank you, we realise this was not clear enough. We have added your suggestion, and we also updated the text to state that the 73 deletion sites span 45 distinct regions (Page 13, line 2), and those are the ones being evaluated (this was specified in the methods, but not clear from the main text).

15. Fig 8. In panel (b), the x axis is not labeled (or explained).

We've added x and y axis labels in this figure and explained the x axis.

16. Methods. Page 17, line 21. I'm not sure this paragraph and some of the following is exactly right.

Specifically, on line 22, you say that "S is non-empty and totally ordered", but you don't specify what this total order is. If you intend to use the partial order defined by the edges of the DAG itself, then I don't believe $c(v)$ has to be unique (unless the DAG is a lattice, which you have not specified).

We realise we did not explain clearly why S is totally ordered. We have therefore updated the methods to make this clear:

<Page 19, line 40>“Let s be the sink of G . Given any opening node v , let S be the set of nodes that are in every path from v to s , excluding v itself. Then S is non-empty because s belongs to S . Let a and b be any elements of S . Then, by definition of S , there exists a path that contains both a and b . Therefore, using the partial order defined by the edges of G , a and b are comparable and it follows that S is a totally ordered finite non-empty set.”

We have also added an illustration of S in Supplementary Fig. 2 to make this point clear.

After thinking about this a bit, I think what you are intending here is that gramtools supports DAGs such that there exists a total order T that is a topological sort of the DAG and that the DAG must have certain properties with respect to T , in particular that T is a balanced-bracket ordering of the DAG.

On top of the clarification of S , we have rewritten the definitions section of the paper considerably to make it clear how NDAGs are defined:

<Page 20, line 9> “Definition Let G be a DAG with a unique source and unique sink. A variant site

is defined as the subgraph of G induced from $\{u, c(u)\}$ or from $\{o(v), v\}$, where u is any opening node and v is any closing node of G .

We remind the reader that for any DAG G , there exists at least one ordering of all the nodes v_0, v_1, \dots, v_n such that given any edge (v_i, v_j) of G , v_i appears before v_j in the ordering. This is called a topological ordering of G . Using the above definitions we can now define the type of graph that is supported by gramtools, which we call a “nested directed acyclic graph”.

Definition Let G be a DAG with a unique source and unique sink. G is said to be a nested directed acyclic graph (NDAG) if there exists a topological ordering of all nodes v_0, v_1, \dots, v_n such that adding brackets to this ordered list of nodes according to the following rules results in balanced opening and closing brackets:

- 1 For each opening node u , add $[$ u after u , and add $]$ u before $c(u)$;
- 2 For each closing node v , add $[$ v after $o(v)$ and add $]$ v before v , unless these brackets were already added by case 1.”

We believe this addresses your comment, in specifying that indeed an NDAG is a DAG such that a topological sort of the DAG with added brackets for opening/closing nodes has balanced brackets.

If you reframe it this way, then it is also important to distinguish between whether \$T\$ exists and whether you have an algorithm to find a suitable total order \$T\$ if it exists.

We agree with you in that we did not provide an algorithm to check if a DAG is an NDAG. Instead of doing this, we use a constructive algorithm to generate NDAGs. We apologise that this was not immediately obvious from the submitted manuscript, as the relevant section describing this was a bit further down in the article, not right next to the graph definitions. We now make this clearer by moving the graph construction section to directly after the graph definitions methods section and by expliciting how our RCC algorithm for graph construction (published in [15]) produces an NDAG. We have also added a new supplementary figure (Supp. Fig. 5) to illustrate the algorithm. To summarise here: the RCC algorithm constructs graphs from multiple-sequence alignments (MSA) by directly mimicking a hierarchical bracketing process. Given an MSA, vertical slices are found where all alleles are identical. Regions between these slices correspond to variant sites. For each such region, the alleles are clustered using K-means, and then within clusters the same algorithm is applied recursively. If we assign each variant region/cluster a pair of open/close brackets, it can be easily seen that these will be balanced (see Supp. Fig. 5 for an illustration), and thus the algorithm creates an NDAG.

The question raised by the reviewer, of how to test a random DAG to see if it is an NDAG is an interesting one, but we respectfully suggest it is out of scope for our paper: we show we can construct such a graph directly from an MSA and that is all we need in this paper.

As an aside, although I wasn't able to work this out in detail, I wonder whether the set of NDAGs supported by gramtools are exactly those DAGs that are (finite, and therefore complete) lattices. I wasn't able to sketch a proof (or counter-example) in the margins, but the idea would be that \$c(v)\$ would be the meet of the successors of \$v\$ and \$o(v)\$ would be the

join of the predecessors of v . You would then have to prove that every lattice L has a total order T such that T is a balanced-bracket ordering for L , ideally by showing a way to construct T . And conversely show that every DAG D , if there exists a total order T that is a balanced-bracket ordering for D , then D is a lattice.

We found this question very interesting and will consider it for future investigation. However, we agree that this is an aside, and would argue that this can therefore be left out of this paper.

17. Page 18, line 7: I think you mean all walks from s_1 to s_2 (not "all walks from s_1 "). That is correct and was updated, thank you.

18. Page 23, line 40: Since there are only 5 manual validations, it wouldn't be bad to include all 5 plots in the supplement as you did for the example in Supp Fig 11. We agree and added them in (now Supplementary Figures 12-16).

19. Supp Fig 12 does not seem to be referenced from the text. Thank you for spotting this, we have added a reference to this Figure on Page 13, line 32 (now Supp. Fig. 18).

20. Supplement

It is a little weird to have the jVCF specification stuck into the middle of the supplement. The reader has to scroll through it to find the figures referenced by the manuscript. Does it really need to be in the supplement if it is available online? If it is attached as a supplement, I would suggest making it a different supplementary file or putting it at the end of the supplement in a separate section.

We agree and have moved the jVCF specification to the end of the supplementary materials. It is, as you mention, also available online.

Reviewer #3: ===

In this paper by Letcher et al. a novel approach called gramtools is introduced to model genomes are directed acyclic graph. They proposed approach allowed successive hierarchical subgraphs which gives the ability to genotype and represent regions of genomes with high variability between samples.

Major Comments:

1. The results in Figure 7 is quite good. However, I think the paper would benefit from more discussion and breakdown of their genotyping ability especially based on different types of variants (and or size of events)

Thank you for this suggestion. We broke down variant calls by type (insertion, deletion, SNPs) and size (1-10, 11-50, 51 and above) and evaluated genotyping performance in each of these categories. This allowed us to identify which variant categories each tool performed more or less well on. We have added a discussion of these results in that section:

<Page 13, line 34>“To understand genotyping performance in more detail, we broke down called variants into different types (insertions, deletions, SNPs) and sizes, and measured precision (what proportion of calls made were correct) and recall (what proportion of the expected calls were recovered) (see Methods). Compared to the other tools, we found vg has a larger number of incorrect and missing small variants (insertions and deletions <10bp, and SNPs). Notably, SNP recall and precision were 57.2% and 86.7% for vg, compared to 91.3% and 93.8% for gramtools and 90.0% and 99.1% for GraphTyper2 (Supp. Fig. 19). Similarly, we found GraphTyper2 has a larger number of incorrect and missed large insertions and deletion calls (>50bp): for large deletions, GraphTyper2 recall and precision were 67.3% and 64.4%, compared to 97.8% and 99.6% for gramtools, and 97.1% and 99.5% for vg (Supp. Fig. 19). gramtools achieved the highest recall across all variant categories, but has lower precision than vg or GraphTyper2 for some categories, notably SNPs and small (1-10bp) and mid-size (11-50bp) insertions (Supp. Fig. 19).”

2. Although the authors argue their method is not designed for studying human genome (long genome and low variability), however, I still think some sort of results with human genomes would be interesting (even if only for one chromosome such as chr22).

We do agree this would be an interesting analysis, we do not think it belongs in this paper, for two reasons. First, it is out of scope - this paper is about showing how a hierarchical DAG structure allows us to both model key concepts which we care about (“alternate alleles”) and which are missing from very general genome graphs, and how this solves two important problems for cohort studies: SNPs on alternate alleles, and SNPs “under” deletions. Second, although the wider question about how gramtools and the underlying Burrows-Wheeler index behave on the human genome is interesting, it belongs in a much broader paper about performance limitations and constraints on the various different pan-genome implementations. Genome graph algorithms and implementations are dramatically impacted by 4 independent axes: genome size, genetic diversity, repeat content, and indexing methodology. It is really quite complicated to work through how they impact performance. For these reasons, we feel we cannot add a human analysis to this paper.

3. I think the authors can discuss if the proposed approach can be incorporate with imputation techniques.

This approach, and other related ones, can certainly be integrated with imputation techniques. This can be seen from a couple of perspectives. At a simple level, it is possible to index known haplotypes through graphs (eg see Siren et al <https://academic.oup.com/bioinformatics/article/36/2/400/5538990>) and to use them to inform mapping (<https://www.biorxiv.org/content/10.1101/2020.12.04.412486v2>). These are not exactly what is normally meant by imputation methods in human genomics, where known haplotypes based on sparse markers (eg from a genotyping array, or from a cohort study such as the 1000 genomes) are used to impute SNP genotypes on other samples. These methods and ideas could surely be ported to this graph context, but the multiallelic nature of the sites in the graph would preclude a direct application of existing tools.

Second round of review

Reviewer 2

The authors of addressed all of my comments and concerns.

Reviewer 3

The reviewers have addressed all my comments. I have no further comment.